# Application of an analytical framework for multivariate mediation analysis of environmental data

Max T. Aung[1], Yanyi Song[1], Kelly K. Ferguson [2], David E. Cantonwine[3], Lixia Zeng[4], Thomas F. McElrath[3], Subramaniam Pennathur[4,5,6], John D. Meeker[7] & Bhramar Mukherjee [1,8 ✉]

Diverse toxicological mechanisms may mediate the impact of environmental toxicants (phthalates, phenols, polycyclic aromatic hydrocarbons, and metals) on pregnancy outcomes. In this study, we introduce an analytical framework for multivariate mediation analysis to identify mediation pathways ($q = 61$ mediators) in the relationship between environmental toxicants ($p = 38$ analytes) and gestational age at delivery. Our analytical framework includes: (1) conducting pairwise mediation for unique exposure-mediator combinations, (2) exposure dimension reduction by estimating environmental risk scores, and (3) multivariate mediator analysis using either Bayesian shrinkage mediation analysis, population value decomposition, or mediation pathway penalization. Dimension reduction demonstrates that a one-unit increase in phthalate risk score is associated with a total effect of 1.07 lower gestational age (in weeks) at delivery (95% confidence interval: 0.48–1.67) and eicosanoids from the cytochrome p450 pathway mediated 26% of this effect (95% confidence interval: 4–63%). Eicosanoid products derived from the cytochrome p450 pathway may be important mediators of phthalate toxicity.

[1] Department of Biostatistics, University of Michigan School of Public Health, Ann Arbor, MI, US. [2] Epidemiology Branch, National Institute of Environmental Health Sciences, Research Triangle Park, NC Chapel Hill, US. [3] Division of Maternal and Fetal Medicine, Brigham and Women's Hospital, Harvard Medical School, Boston, MA, US. [4] Department of Internal Medicine-Nephrology, University of Michigan, Ann Arbor, MI, US. [5] Michigan Regional Comprehensive Metabolomics Resource Core, University of Michigan, Ann Arbor, MI, US. [6] Department of Molecular and Integrative Physiology, University of Michigan, Ann Arbor, MI, US. [7] Department of Environmental Health Sciences, University of Michigan School of Public Health, Ann Arbor, MI, US. [8] Department of Epidemiology, University of Michigan School of Public Health, Ann Arbor, MI, US. ✉email: bhramar@umich.edu

Globally, humans are susceptible to widespread exposure to environmental contamination through several anthropogenic operations, including the manufacturing of industrial chemicals, natural resource extraction, and fossil fuel combustion. Pregnant women and developing fetuses are uniquely vulnerable to environmental exposures. The prenatal exposome encompasses the totality of environmental exposures that occur during preconception and pregnancy. Among the vast landscape of pollutants that exist in modern society, we focus the attention of this present study on four classes of environmental contaminants: phthalates, phenols, polycyclic aromatic hydrocarbons, and trace metals. Phthalates are high-production volume chemicals used as plasticizers in numerous consumer products such as vinyl flooring, children's toys, and food packaging[1]. Phenol derived compounds are also widely used in plastics, as well as personal care and pharmaceutical products[2,3]. Polycyclic aromatic hydrocarbons commonly enter the environment through industrial and natural processes, such as the combustion of coal, fossil fuel, and organic debris[4]. Additional anthropogenic activities such as natural resource extraction and electronic waste recycling can yield environmental contamination of heavy metals (e.g., lead, cadmium, mercury)[5]. Biomonitoring analytes derived from these exposure classes can capture the magnitude of exposure in humans to inform health studies.

Exposures to environmental toxicants are suspected risk factors for adverse pregnancy outcomes such as preterm delivery or altered gestational duration[6]. Based on extensive in vitro and animal models, there is evidence indicating inducible receptor activity attributable to exposure classes (e.g., phthalates and phenols with the estrogen receptor, polycyclic aromatic hydrocarbons and the aryl hydrocarbon receptor, toxic heavy metals and NF-κB)[7,8]. Perturbations in receptor signaling and transcription factor activity can result in the altered production of signaling molecules responsible for inflammation and metabolism. Therefore, biomarker signals of toxicant exposures can help disentangle the mediating biological pathways linking toxicant mixtures to pregnancy outcomes, which is critical for early detection of disease and prevention. Mediation pathways can also inform causal inference in human observational studies and thus the need for policy-driven exposure reduction and prevention measures.

In previous analyses in the LIFECODES cohort, we have found that biomarkers of inflammation (cytokines and c-reactive protein [CRP]) and oxidative stress (8-isoprostane) have been associated with increased risk of preterm birth[9,10]. Additionally, we learned that cytochrome p450 and lipoxygenase-derived eicosanoids had the greatest relative predictive capacity for classifying preterm birth compared to a large panel of endogenous biomarkers[11]. Eicosanoids are potent signaling molecules derived from parent fatty acid compounds such as arachidonic acid and linoleic acid[12]. For example, epoxyeicosatrienoic acids (EETs) produced through the cytochrome p450 pathways are involved in vascular remodeling and regulation of inflammation[13]. Therefore these eicosanoids may be up-regulated in disease states involving cardiovascular irregularities, an important precursor for adverse pregnancy outcomes[14]. Among the environmental exposures, we also showed that phthalates have been associated with increased risk for preterm delivery and reduced gestational age at delivery within the LIFECODES cohort[15,16]. Based on these previous studies, there is circumstantial evidence of potential mediation between prenatal phthalate exposure and pregnancy outcomes through eicosanoids.

Precise risk profiles can be estimated through various inferential tools, including the implementation of causal mediation analysis. Early methodological development of mediation analysis enabled investigators to decompose the relationship between a predictor and an outcome, and test whether the total effect is partially mediated by an intermediate variable[17,18]. With the expansion of technology to measure increasingly high-dimensional endogenous biomarkers, the mediation framework needs to evolve in order to handle the analysis of multiple mediators simultaneously. Theoretical advancements in the mediation analysis framework have afforded new applications of mediation analysis to account for multiple mediators. For example, Chen and colleagues have developed methodological applications for leveraging population value decomposition to reduce the mediator matrix and analyze multiple mediators simultaneously[19]. More recently, Song and colleagues developed a Bayesian approach for continuous shrinkage estimation of multiple predictors to identify the most important mediation pathways[20]. In another study, Zhang and colleagues sought to approach this high-dimensional mediation scenario through penalized regularization using the minimax concave penalty and joint significance testing[21]. Another regularization approach that has been applied is a lasso penalty on mediating pathways[22]. Altogether, these statistical methods have pushed the frontier of mediation analysis and created opportunities to make critical scientific discoveries using biomarkers from complex biological pathways.

Currently, there is a critical knowledge gap for addressing a multivariate mediation setting where there are not only high-dimensional mediators, but also multiple toxicants with inherent collinearity and grouping structures. Our study aims to advance methodological applications to investigate a mixtures mediation setting with multiple exposure classes and groups of endogenous biomarker mediators. We propose an analytical framework (Fig. 1) that integrates our previous methodological contributions in environmental risk score construction[16,23,24] with the aforementioned multivariate mediation analysis methods[19,21]. In this framework, we provide a guided discussion on approaches that include one-at-a-time pairwise mediation, exposure dimension reduction, and mediator shrinkage, dimension reduction, and penalization.

## Results

The weighted characteristics of our study sample are reported in Table 1. Our study sample was predominantly White (65.7%) and college-educated (79.2%). The weighted median age (in years) of participants was 32.8 (interquartile range [IQR]: 4.8). The weighted median gestational age at delivery (in weeks) across the study sample was 38.7 (IQR: 2.0). Overall, a majority (51.6%) of participants had a BMI < 25 kg/m$^2$ at their initial visit. There were very few participants who smoked (5.4%) or drank alcohol (5.0%) during pregnancy. Sample characteristics were also compared to the larger parent LIFECODES cohort in Supplemental Table 1. Some demographic and health variables are relatively close in proportion to the larger LIFECODES cohort (within 5% difference for any single category): BMI, tobacco use, alcohol use (Supplemental Table 1). However, maternal race, education, and health insurance provider were trending toward greater white and socioeconomic status in the current subset sample compared to the larger LIFECODES cohort (Supplemental Table 1).

**Associations between exposures and outcomes.** Coefficients for total effects from single pollutant associations with overall preterm birth are presented in Supplementary Data 1A. Coefficients for spontaneous preterm birth are presented in Supplementary Data 1B. A forest plot of total effects for gestational age at delivery are presented in Fig. 2. Given the limited power in sample size for this study, we observed positive associations between phthalate metabolites (Mono-2-ethyl-hexyl phthalate [MEHP],

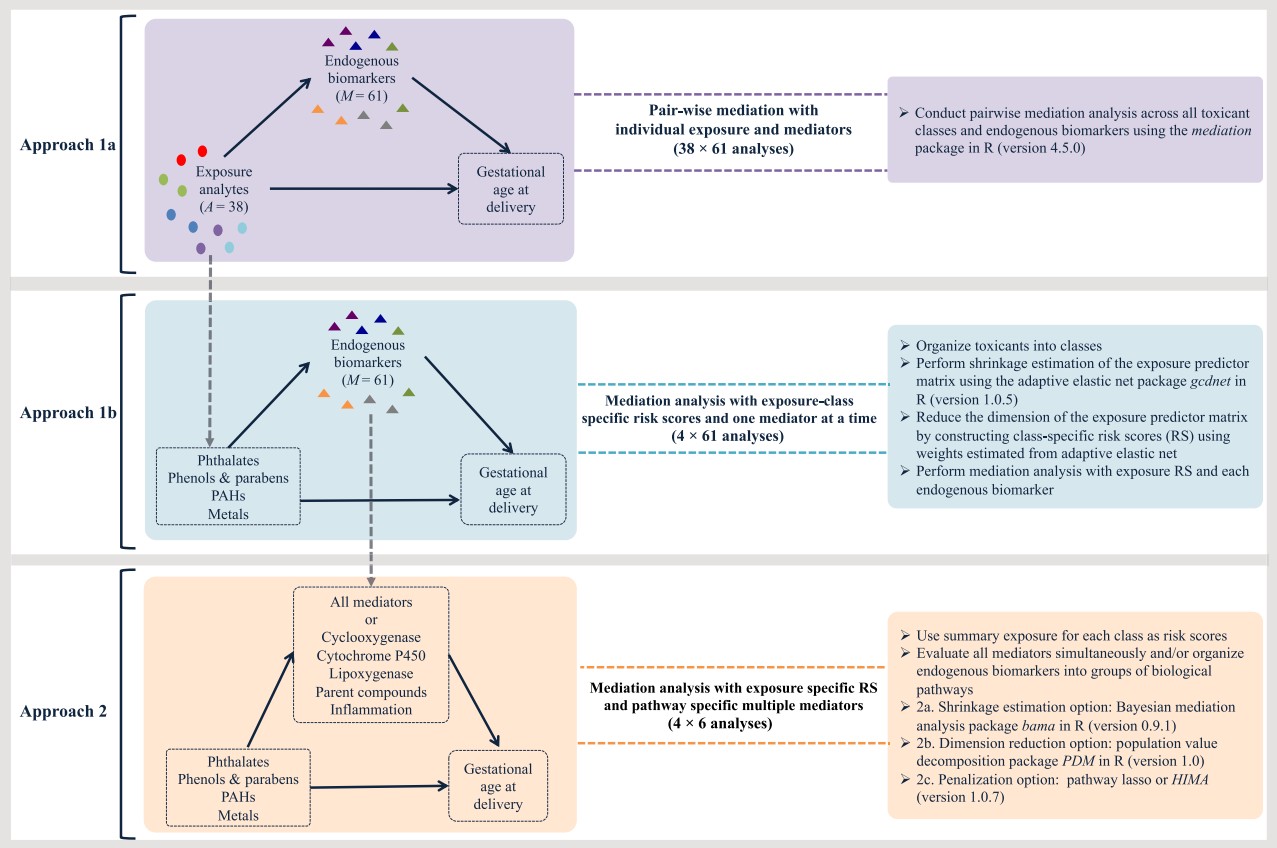

**Fig. 1 Analytical framework for conducting multivariate mediation analysis of toxicant mixtures, endogenous biomarkers, and birth outcomes.** Approach 1a describes pair-wise mediation analysis of single exposures and mediators. Approach 1b describes methodology for conducting exposure dimension reduction followed by pair-wise mediation analysis. Approach 2 describes methodology to approach mediator matrix shrinkage, dimension reduction, or penalization.

mono-2-ethyl-5-hydroxyhexyl phthalate [MEHHP], mono-2-ethyl-5-oxohexyl phthalate [MEOHP], and mono-2-ethyl-5-carboxypentyl phthalate [MECPP]) and odds of overall preterm birth and spontaneous preterm birth, however, these estimates were not significantly different from the null ($p > 0.05$). Associations with phthalates have been previously explored in the larger LIFECODES cohort, where several of these associations were significant[15,16]. Three toxicants (2,4-dichlorophenol [2,4-DCP], 1-hydroxyphenanthrene [1-PHE], and thallium [Tl]) were associated with decreased odds of overall preterm birth (Supplementary Data 1A). When focusing on spontaneous preterm birth, four toxicants (2,4-DCP, 2,5-dichlorophenol [2,5-DCP], triclosan [TCS], and 1-PHE) were associated with decreased odds (Supplementary Data 1B). Similar directions of associations were observed in a previous analysis of these phenols in the larger LIFECODES cohort as well[25]. Modeling gestational age at delivery revealed three phthalate metabolites (MEHHP, MEOHP, and MECPP) associated with decreased gestational age at delivery (Fig. 2). 2,5-DCP was associated with increased gestational age at delivery (Fig. 2).

**Pairwise mediation analysis.** Estimated total, natural direct, and natural indirect effects for each pair of toxicants and mediators are presented for models of overall preterm birth (Supplementary Data 1A), spontaneous preterm birth (Supplementary Data 1B), and gestational age at delivery (Supplementary Data 1C). For overall preterm birth, we observed 31 natural indirect effects ($p < 0.05$), although those estimates were not robust to false discovery adjustment (q-value > 0.2) (Supplementary Data 1A). The

mediators that appeared the most among these mediation effects were resolvin D1 [RVD1] and 11,12-dihydroxy-eicosatrienoic acid [11,12-DHET] (Supplementary Data 1A). There were 72 natural indirect effects ($p < 0.05$; q-value > 0.2) that were observed for spontaneous preterm birth (Supplementary Data 1B). These natural indirect effects did not have corresponding significant total or direct effects, indicating incomplete mediation. Last, there were 24 natural indirect effects ($p < 0.05$; q-value > 0.2) observed for gestational age at delivery (Fig. 3 and Supplementary Data 1C). Among those mediation pathways for gestational age at delivery, one pair (MEHHP and 5(6)-EET) had a positive mediation signal but also exhibited significant negative total and direct effects (Supplementary Data 1C).

**Multivariate mediator analysis.** Bayesian continuous mediation shrinkage and population value decomposition for mediator dimension reduction were both restricted to the continuous outcome variable gestational age at delivery. When we conducted mediator shrinkage, mediation pathway signals were sparse and posterior inclusion probabilities did not exceed 0.2 (Supplementary Data 2). Mediator dimension reduction allowed for evaluation of all mediators simultaneously and whole groups of mediators as potential mediation pathways for individual toxicants. In this approach, we observed 15 natural indirect effects ($p < 0.05$; q-value > 0.2) (Fig. 4 and Supplementary Data 3). MECPP in particular had a positive indirect effect ($p < 0.05$; q-value > 0.2) and positive total effect ($p < 0.05$; q-value > 0.2), however the direction of the natural indirect effect was negative—indicating inconsistent mediation (Supplementary Data 3). To

**Table 1 LIFECODES Prospective pregnancy cohort profile, weighted for inverse probability weights of case-control sampling (N = 161).**

| Sample characteristics | |
|---|---|
| | **Median (IQR)** |
| Age (years) | 32.8 (4.8) |
| Gestational age at delivery (weeks) | 38.7 (2.0) |
| | **Count (%)** |
| Overall preterm birth | |
| Case | 52 (32.3%) |
| Control | 109 (67.7%) |
| Spontaneous preterm birth | |
| Case | 30 (21.6%) |
| Control | 109 (78.4%) |
| Initial visit BMI (median 10 weeks gestation) | |
| <25 kg/m$^2$ | 83 (51.6%) |
| 25–29.9 kg/m$^2$ | 46 (28.7%) |
| ≥30 kg/m$^2$ | 32 (19.7%) |
| Race | |
| White | 106 (65.7%) |
| Black | 17 (10.9%) |
| Other | 38 (23.4%) |
| Education level | |
| High school degree | 18 (11.5%) |
| Technical school | 15 (9.3%) |
| Junior college or some college | 56 (34.6%) |
| College graduate | 72 (44.6%) |
| Insurance | |
| Private/HMO/Self-pay | 146 (90.7%) |
| Medicaid/SSI/MassHealth | 15 (9.3%) |
| Tobacco use | |
| No smoking during pregnancy | 152 (94.6%) |
| Smoked during pregnancy | 9 (5.4%) |
| Alcohol use | |
| No alcohol use during pregnancy | 153 (95.0%) |
| Alcohol use during pregnancy | 8 (5.0%) |
| Fetal sex | |
| Female | 75 (46.3%) |
| Male | 86 (53.7%) |

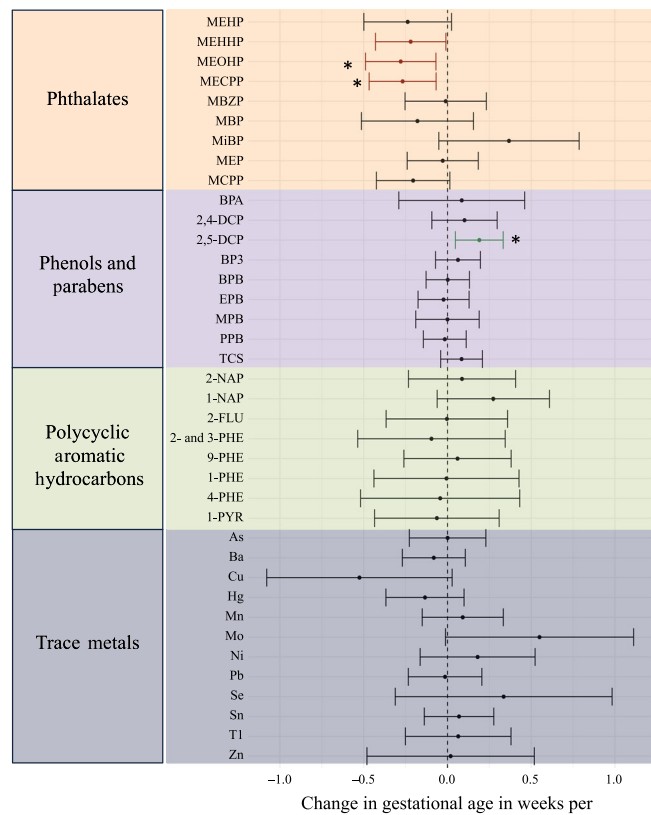

**Fig. 2 Associations between individual toxicants and gestational age at delivery (in weeks) estimated by weighted multiple linear regression (N=161).** Models adjusted for specific gravity, maternal age, race, education, health insurance, and BMI at the initial study visit. Toxicants are organized on the y-axis and color-coded to differentiate classes: orange (phthalates), purple (phenols and parabens), green (polycyclic aromatic hydrocarbons), and gray (trace metals). Data are presented as point estimates (linear regression beta coefficients) and 95% confidence intervals (beta coefficient * ±1.96 (standard error[beta coefficient]). All p-values estimated using a two-sided test for t-statistics from linear regression models. Adjustments for multiple comparisons were accounted for by calculating q-values. *q-values = 0.12.

investigate the contributions of individual mediators towards the direction of mediation constructed for all mediators simultaneously, we illustrated correlation coefficients between the direction of mediation and individual mediators in Supplemental Fig. 1. Only two mediators, RVD1 (lipoxygenase pathway) and 11,12-DHET (cytochrome p450 pathway) exhibited Pearson correlation coefficients greater than |0.3| with the direction of mediation constructed for all mediators simultaneously, indicating that these two pathways may have a larger influence in this dimensionally reduced mediator vector.

**Exposure and mediator dimension reduction.** Continuing to focus strictly on gestational age at delivery, we reduced the dimensionality of the exposure matrix by creating environmental risk scores for each exposure class: phthalates, phenols, metals, and polycyclic aromatic hydrocarbons. Prior to exposure dimension reduction, individual toxicants had high within-class correlation, with the highest correlation coefficient (ρ = 0.97) observed between the phthalate metabolites MEOHP and MEHHP (Supplemental Fig. 2 and Supplementary Data 4). Creating environmental risk scores removed within-class correlations and we observed low correlation coefficients between risk scores (ρ ranging from |0.03 to 0.26|) (Supplemental Table 2). Mediation analysis of environmental risk scores and single

mediators did not reveal any significant mediation signals (Supplementary Data 5).

Figure 5 illustrates forest plots of total effects, direct effects, and natural indirect effects for each environmental risk score across each mediator group (All biomarkers, cyclooxygenase pathway, cytochrome p450 pathway, lipoxygenase pathway, parent lipid compounds, and inflammatory biomarkers). Oxidative stress biomarkers and protein damage biomarkers were excluded in the subset grouping because there were less than five biomarkers in both of these groups. The inflammatory markers group effect could not be estimated by population value decomposition, indicating either sparse or null mediation signals across all four environmental risk scores. The fact that some mediator group effects were not estimated by the population value decomposition algorithm is partly due to the bivariate correlation structure between exposures and mediators. We observed a low correlation (ρ < 0.3) between almost all of the exposures and inflammatory markers, with the exception of one phthalate metabolite MECPP and the cytokine TNF-α (ρ = 0.45) (Supplementary Data 4). Similarly, the parent compound mediator group effect could not be estimated by population value decomposition for the metal risk

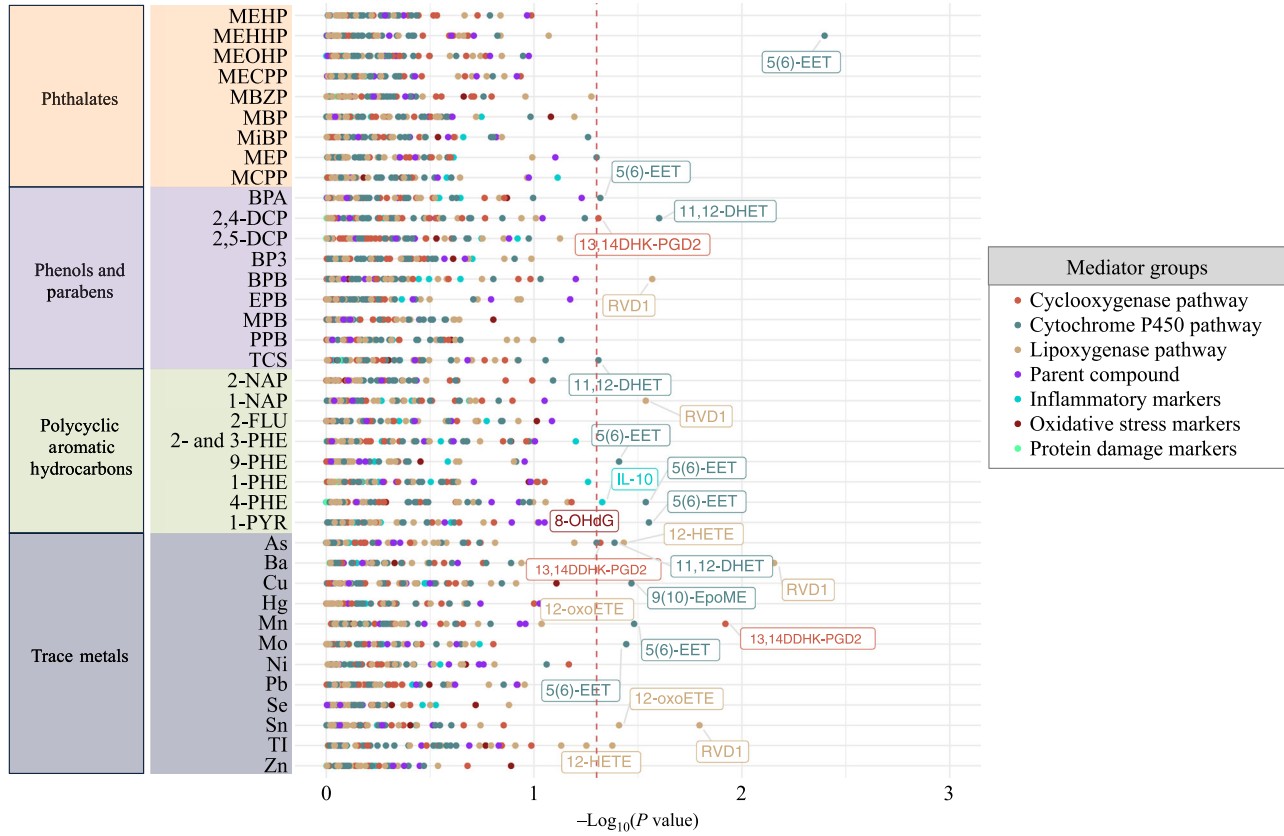

**Fig. 3 Estimated –log₁₀(p-values) of natural indirect effects from pairwise weighted mediation models for gestational age at delivery (N = 161).** Dashed vertical red line signifies *p*-value threshold of 0.05, and labeled point estimates are below that threshold. Models adjusted for specific gravity, maternal age, race, education, health insurance, and BMI at the initial study visit. Toxicants are organized on the y-axis and color-coded to differentiate classes: orange (phthalates), purple (phenols and parabens), green (polycyclic aromatic hydrocarbons), and gray (trace metals). All p-values estimated using a two-sided test for t-statistics from linear regression models. Adjustments for multiple comparisons were accounted for by calculating q-values. Reported p-value estimates have corresponding q-values exceeding 0.2.

score. This is also consistent with observations of low correlations $(-0.18 < \rho < 0.17)$ between metals and parent compounds (Supplementary Data 4).

The phthalate risk score exhibited natural indirect effects when evaluating all biomarkers simultaneously, and grouped analysis revealed that the mediation effect was chiefly driven by the cytochrome p450 pathway (Fig. 5). One unit increase in the phthalate risk score was associated with a total effect of 1.07 lower gestational age (in weeks) at delivery (95% confidence interval: 0.48–1.67) (Fig. 5 and Supplementary Data 6). From this total effect, 26% (95% confidence interval: 4-63%) of the effect was mediated through the cytochrome p450 pathway (Fig. 5 and Supplementary Data 6).

The phenol risk score exhibited only significant positive total and direct effects in association with gestational age at delivery in the grouped analyses for cyclooxygenase pathway, cytochrome p450 pathway, lipoxygenase pathway, and parent compounds (Fig. 5). The corresponding mediation effects were not significant. When evaluating all biomarkers simultaneously, the phenol risk score's total, and direct effects were no longer significant, albeit these effects alongside the mediation effect were suggestive (Fig. 5 and Supplementary Data 6). A plausible explanation lies in the fact that analysis of all biomarkers simultaneously, includes inflammatory, protein damage, and oxidative stress markers, which collectively diminished the total and direct effects observed in the subgroup analysis of eicosanoid enzymatic pathway groups and parent compounds.

**Comparative methods**. Table 2 reports mediators that remained for each risk score after applying pathway lasso shrinkage. We observed that select mediators from the cytochrome p450 pathway (17-hydroxy-eicosatetraenoic acid [17-HETE], 18-hydroxy-eicosatetraenoic acid [18-HETE]), lipoxygenase pathway (Leukotriene B4 [LTB4]), parent lipid compounds (Docosahexaenoic acid [DHA]), and inflammation (CRP, interleukin-1β [IL-1β], interleukin-6 [IL-6], interleukin-10 [IL-10], tumor necrosis factor-α [TNF-α]) groups remained in the relationship between the phthalate risk score and gestational at delivery (Table 2). For the metal risk score, pathway lasso selected mediators in the cyclooxygenase pathway (Prostaglandin B2 [PGB2], prostaglandin A2 [PGA2], prostaglandin E1 [PGE1], and bicycle-prostaglandin E2 [BCPGE2]), cytochrome p450 pathway (17-HETE, 18-HETE, and 11,12-DHET), inflammation (IL-6), and protein damage group (3-chlorotyrosine [CY]) (Table 2). For the phenol risk score, pathway lasso selected mediators from the cytochrome p450 pathway (17-HETE, 9,10-dihydroxy-octadece-noic acid [9,10-DiHOME], and 8,9-epoxy-eicsatrienoic acid [8 (9)-EET]), lipoxygenase pathway (13-oxooctadeca-dienoic acid [13-oxoODE]), inflammation [CRP], and oxidative stress group (8-hydroxydeoxyguanosine [8-OHdG] and 8-isoprostane [8-IP]) (Table 2). The polycyclic aromatic hydrocarbon risk score was linked with two mediators, from the cyclooxygenase pathway (PGE1), and oxidative stress group (8-IP) (Table 2).

In Table 3, we document the findings from the *hima* function that applies a minimax concave penalty and joint significance test

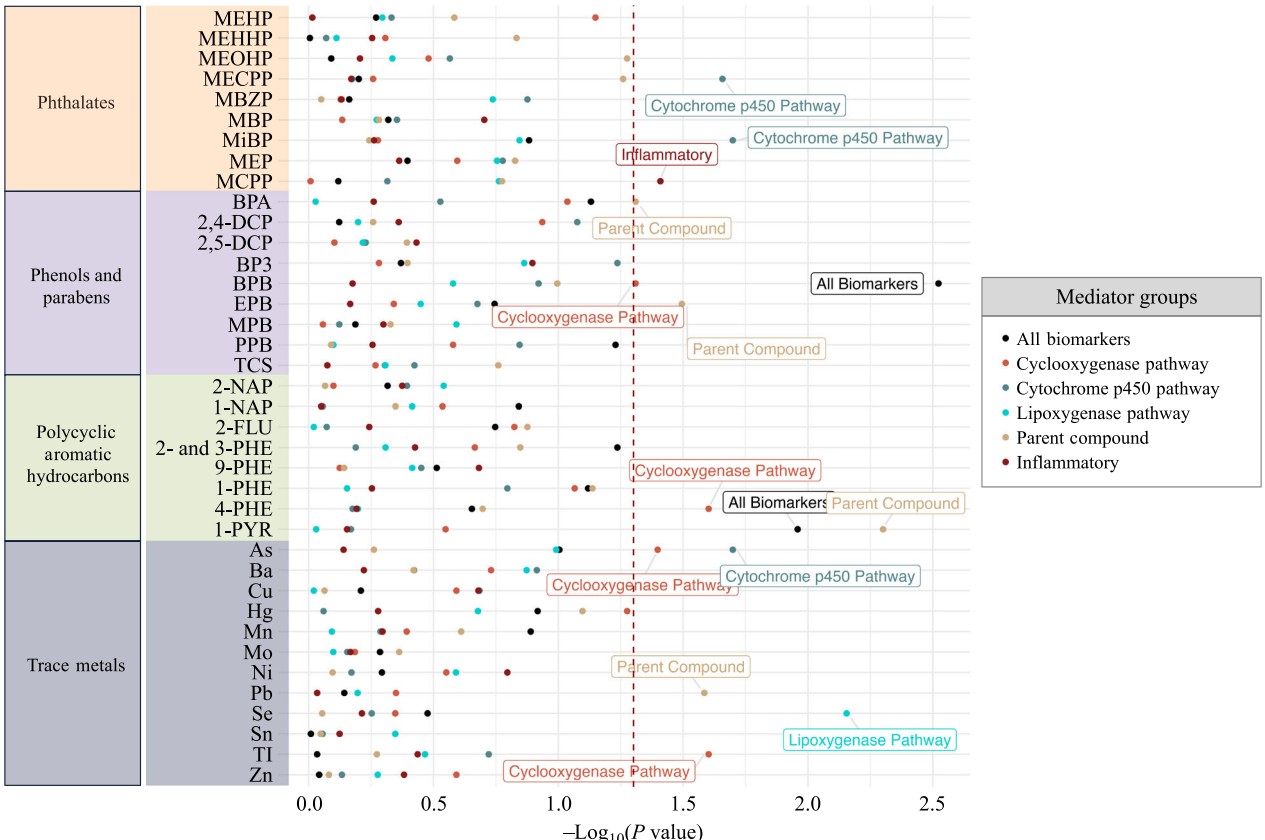

**Fig. 4 Estimated −log$_{10}$(p-values) of natural indirect effects from pairwise mediation models for gestational age at delivery (N = 161) using mediator group effects derived from population value decomposition.** Mediator groups with less than five biomarkers (oxidative stress and protein damage) were omitted from population value decomposition. Dashed vertical red line signifies p-value threshold of 0.05, and labeled point estimates are below that threshold. Models adjusted for specific gravity, maternal age, race, and BMI at the initial study visit. Toxicants are organized on the y-axis and color-coded to differentiate classes: orange (phthalates), purple (phenols and parabens), green (polycyclic aromatic hydrocarbons), and gray (trace metals). All p-values estimated using a two-sided test for t-statistics from linear regression models. Adjustments for multiple comparisons were accounted for by calculating q-values. Reported p-value estimates have corresponding q-values exceeding 0.2.

for mediator pathways. We observed that in this method, the phthalate risk score was significantly linked (q-value < 0.1) to IL-10 (inflammation) (Table 3). Although not meeting the FDR threshold, select cytochrome p450 products (20-hydroxy-eicosa-tetraenoic acid [20-HETE], 20-carboxy arachidonic acid [CAA], and 11,12-DHET) also showed subtle mediation signals for the phthalate risk score (Table 3). The phenol risk score was linked (FDR q-value < 0.1) to IL-10 (Table 3). Last, the polycyclic aromatic hydrocarbon risk score was also linked (FDR q-value < 0.2) to the oxidative stress mediator 8-OHdG (Table 3).

An alternative approach to population value decomposition derived mediator group effects is the use of sparse principal component constructed mediators. This approach assumes sparse effects on the outcome variable among individual mediators. Sparse principal component mediation analysis yielded 24 principal components that accounted for 80% of the variance in the mediator matrix. There was no single sparse principal component that exhibited a significant mediation signal between the phthalate risk score and gestational age at delivery, however, a majority (66%) of the components had natural indirect effect estimates in the same direction as what we observed when evaluating all biomarkers simultaneously using population value decomposition (Supplementary Data 7). When evaluating contributions in loading factors for each sparse principal component, we observed that mediators from the cytochrome p450 pathway had the greatest number of loadings above 0.3 compared to other mediator groups (Supplemental Fig. 3).

**Sensitivity analysis.** We assessed the sensitivity of our final mediation model for phthalate risk score and all mediators simultaneously using the *medsens* function in the mediation package. This sensitivity analysis revealed that in order to diminish the natural indirect effect, there would need to be unmeasured confounders in the respective mediator and outcome models such that the product of the variance explained ($\tilde{R}^2$) by those confounders equals to 0.187 (Supplemental Fig. 4). We further assessed the sensitivity of our final mediation model by calculating the E-value, which is an estimate of the magnitude of association required by unmeasured confounders to diminish the mediation signal we observed in our study. The E-value for our final model was 1.55 (lower bound 1.15) (Supplemental Fig. 5). Therefore, an unmeasured confounder beyond the variables adjusted in our final model would need to have a risk ratio of 1.55 (lower bound 1.15) in association with gestational age at delivery and the mediator matrix in order to completely diminish the mediation signal that we observed in our study.

## Discussion

Conducting high-dimensional mediation analysis of toxicant mixtures and endogenous biomarkers advances the scientific discovery of intermediate pathways by which environmental contamination may impact pregnancy outcomes. Furthermore, mediation analysis enhances causal inference in observational studies. In this study, we designed a framework to guide

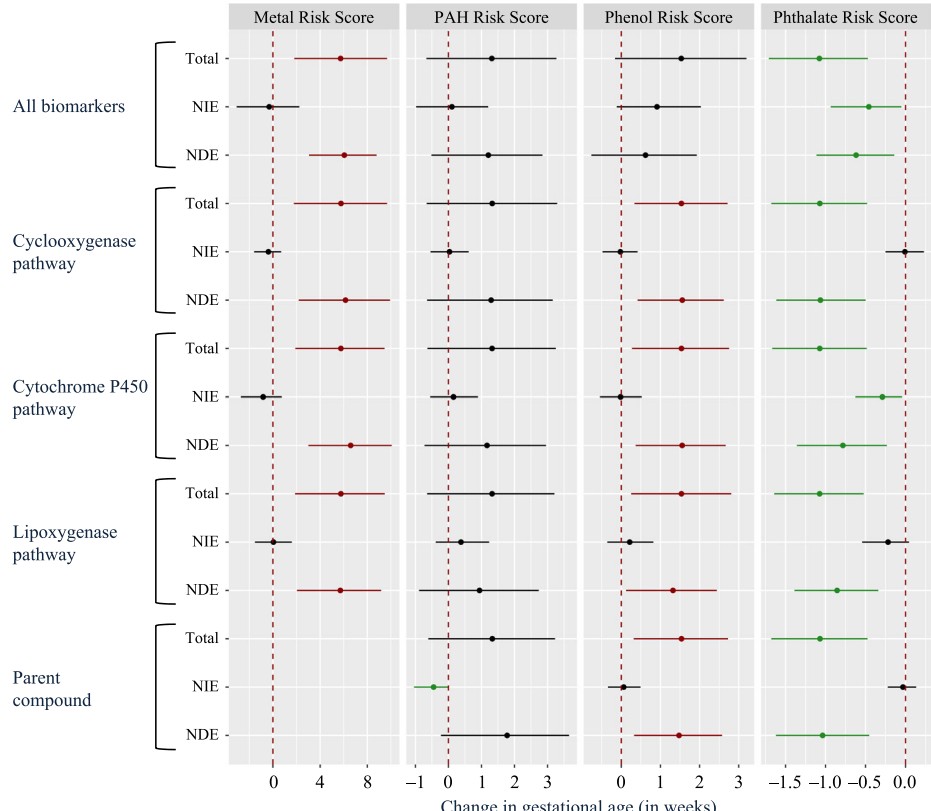

**Fig. 5 Forest plot of mediation analysis for environmental risk scores, mediator group effects, and gestational age at delivery ($N = 161$).** Models adjusted for specific gravity, maternal age, race, and BMI at the initial study visit. All p-values estimated using a two-sided test for t-statistics from linear regression models. Adjustments for multiple comparisons were accounted for by calculating q-values. Estimates have corresponding q-values exceeding 0.2. Abbreviations: Natural direct effect (NDE); Natural indirect effect (NIE). In cases where population value decomposition did not produce an estimate of a mediator group effect (parent compounds and metal risk score as well as the inflammatory markers group effect for all risk scores), the results are absent from the Figure. This indicates that the mediation signal was sparse for those particular pairs of exposures and mediators. Data are presented as point estimates (linear regression beta coefficients) and 95% confidence intervals (beta coefficient * ±1.96(standard error[beta coefficient]).

---

**Table 2 Pairs of environmental risk scores and mediators selected by pathway lasso.**

|  | Metal risk score | PAH risk score | Phenol risk score | Phthalate risk score |
|---|---|---|---|---|
| Cyclooxygenase Pathway | PGB2, PGA2, PGE1, BCPGE2 | PGE1 | — | — |
| Cytochrome P450 Pathway | 17-HETE, 18-HETE, 11,12-DHET | — | 17-HETE, 9,10-DiHOME, 8(9)-EET | 17-HETE, 18-HETE |
| Lipoxygenase Pathway | — | — | 13-oxoODE | LTB4 |
| Parent Compound | — | — | — | DHA |
| Inflammatory | IL-6 | — | CRP | CRP, IL-1β, IL-6, IL-10, TNF-α |
| Oxidative Stress | — | 8-IP | 8-OHdG, 8-IP | — |
| Protein Damage | CY | — | — | — |

Pathway lasso implemented all biomarkers simultaneously, therefore the categorization of individual mediators into mediator groups are chiefly for biological relevance not statistically unique models.

---

researchers through an important mediation analysis setting that accounts for multiple predictors and mediators. This is particularly applicable for the analysis of high-dimensional data of endogenous omics-scale biomarkers. Our framework introduces single pairwise mediation analysis, and underlines concerns with this approach, including multicollinearity between exposures, and sparsity of signals when evaluating any single biomarker. Ultimately, our framework arrives toward a dimensionally reduced setting with exposure class risk scores and mediator group effects organized by endogenous biomarker groups. Reducing the dimensions of the exposure and mediator matrices can be a powerful tool for causal inference, especially when coupled with

literature derived organization of the mediator matrix by biological pathways.

Pairwise mediation yielded subtle evidence of natural indirect effects, which was expected given the limited sample size and insufficient power to detect fully significant natural direct and total effects. Additionally, due to the correlation structure of individual toxicants and mediators, pairwise mediation models are not completely independent. Nonetheless, the findings from pairwise mediation provide initial evidence of specific toxicant-mediator pairs that may appear in larger study samples. When we reduced the dimensionality of the exposure and mediator matrices, we combined the effects of individual toxicants into

**Table 3 Pairs of environmental risk scores and mediators with detected mediation effects estimated using the minimax concave penalty and joint significance test in *hima*.**

|  | Metal Risk Score | PAH risk score | Phenol risk score | Phthalate risk score |
|---|---|---|---|---|
| Using all biomarkers | IL-10, 8-OHdG, RVD1 | IL-10, 8-OHdG, RVD1 | RVD1 | IL-10[a], 8-OHdG, RVD1 |
| Cyclooxygenase Pathway | 13,14DHK-PGD2, PGD3 | 13,14DHK-PGD2, PGD3 | 13,14DHK-PGD2, PGD3 | 13,14DHK-PGD2, PGD3 |
| Cytochrome P450 Pathway | 20-HETE, 11,12-DHET | 20-HETE, CAA, 11,12-DHET | 20-HETE, CAA, 11,12-DHET | 20-HETE, CAA, 11,12-DHET |
| Lipoxygenase Pathway | RVD1 | RVD1 | RVD1 | RVD1 |
| Parent Compound | AA | AA[a], LA | AA, LA | AA, LA |
| Inflammatory | IL-10 | IL-10 | IL-10[a] | IL-10[a] |

[a]q-values < 0.2 and exact values were as follows: IL-10 and phthalate risk score [q-value1 = 0.02, q-value2 = 0.03], AA and PAH risk score [q-value = 0.18], IL-10 and phenol risk score [q-value = 0.19]
Mediator groups with less than five biomarkers (oxidative stress and protein damage) were omitted from group-specific analysis using *hima*.

their designated classes, therefore estimating their cumulative effect. This approach also combined the mediator contributions into enzymatic pathways or groups. Importantly, the phthalate risk score was associated with decreased gestational age at delivery and this effect was mediated by all of the mediators simultaneously. As we disaggregated the mediators into groups, we learned that this mediation effect was largely driven by cytochrome p450 derived eicosanoids. This finding is also biologically consistent with previous work where we observed that the cytochrome p450 pathway eicosanoids yielded the greatest predictive capacity for preterm birth and spontaneous preterm birth, more so than any of the other mediator groups (cyclooxygenase, lipoxygenase, and parent compound)[26]. We also observed suggestive evidence of mediation with the lipoxygenase pathway in the association between the phthalate risk score and gestational age at delivery. Overall this analytical framework demonstrates the differences observed between single biomarker analysis and multivariate mediator analysis for toxicant mixtures.

Throughout our analytical framework, we explored a sophisticated ensemble of contemporary approaches for high-dimensional mediation analysis: Bayesian continuous shrinkage mediation[20], population value decomposition[19], pathway lasso[22], and application of minimax concave penalty and joint significance testing[21]. The theoretical basis for each of these methods applies unique penalties and estimation algorithms; therefore we expect a substantive degree of non-uniformity in results across methods. Nonetheless, findings that appear repetitively across methods materialize relationships that indicate importance, afford greater confidence of a true mediation pathway, and warrants deeper investigation through replication in independent samples. Chiefly, we learned that the phthalate risk score appears to be linked with the whole cytochrome p450 pathway effect using population value decomposition. When comparing this finding with the other multivariate mediation methods, individual cytochrome p450 metabolites were identified in both the pathway lasso penalty and the minimax concave penalty with joint significance testing. Furthermore, there are suggestive signals of lipoxygenase pathway and metabolites being linked to the phthalate risk score across each of these multivariate mediation methods. Another important finding was the lack of a mediation effect observed between the phthalate risk score and cyclooxygenase pathway or its metabolites when applying each of the high-dimensional mediation methods. Our study provides consistent evidence that phthalates may impact gestational age at delivery mediated through the cytochrome p450 pathway – and to a lesser degree the lipoxygenase pathway.

The cytochrome p450 enzymes are a conserved superfamily of enzymes that function to metabolize environmental toxicants, drugs, and endogenous compounds[27]. Transformation of endogenous fatty acid compounds such as arachidonic acid results in the production of bioactive signaling molecules, which can influence cellular biology through altering membrane permeability and transcription factor activity[28]. For example, cytochrome p450 derived HETEs can influence renal vascular changes and impact other enzymes, such as inhibiting adenosine triphosphatase [ATPase] activity or stimulating peroxisome proliferator-activated receptor-gamma [PPARγ][29,30]. Additionally, EETs can mediate anti-inflammatory effects by inhibiting NF-κB activity and increasing MAP kinase activity, thereby influencing cyclic AMP concentrations[13]. Cytochrome p450 activity is partially stimulated by elevated pro-inflammatory cytokines such as TNF-α[28]. As such, we expect maternal biomarker profiles to reflect observable differences in cytochrome p450 products in disease states. Furthermore, the circulation of cytochrome p450 products is critically important for regulating cardiovascular functions, such as vasodilatory changes and vascular inflammation partly through the expression of endothelial adhesion molecules for chemotaxis and leukocytic migration[28]. The underlying mechanisms associated with reduced gestational age duration and preterm delivery include maternal inflammation and systemic oxidative stress[31,32]. Therefore, the cardiovascular and inflammatory responsiveness of cytochrome p450 activity underlines the critical importance of these enzymes for pregnancy outcomes.

The mediation signal between phthalates and cytochrome p450 pathway underline an important toxicological mechanism. Several phthalate parent compounds (dimethyl phthalate, diethyl phthalate, dipropyl phthalate, dibutyl phthalate, benzyl butyl phthalate, dicyclohexyl phthalate) have been previously shown to inhibit cytochrome p450 enzymatic activity in vitro[33]. However, another in vitro study demonstrated that cytochrome p450 activity is enhanced with di-2-ethylhexyl phthalate exposure, and this relationship depended on both the expression levels of the pregnane X receptor as well as concentrations of glucocorticoids[34]. Given that cytochrome p450 enzymes are involved in the metabolism of xenobiotics, they may be sensitive to phthalate exposures. Furthermore, since cytochrome p450 enzymes also metabolize endogenous signaling molecules, an increased burden in phthalate exposure may impose adverse perturbations in the normal functions of these enzymes.

Studies that have assessed mediation pathways for phthalates in the context of pregnancy outcomes focused on limited mediators and did not assess the whole biomarker groups[35–37]. In the larger LIFECODES cohort, Ferguson and colleagues conducted mediation analysis using the oxidative stress biomarker 8-IP as a potential mediator in the relationship between phthalate metabolites and preterm birth ($n_{\text{cases}} = 130$, $n_{\text{controls}} = 352$)[37]. This study demonstrated that 8-IP mediated the relationship between preterm birth and MEHP, MECPP, and mono-n-butyl phthalate (MBP)[37]. A pilot study based in China ($n = 115$) investigated DNA methylation signatures as potential mediators but did not find evidence of mediation in the relationship between prenatal

phthalates and gestational age[35]. Extending to outcomes beyond the pregnancy, a study based in Norway investigated maternal thyroid hormones as potential mediators in the relationship between prenatal phthalate exposure and attention-deficit hyperactivity disorder ($n_{cases} = 297$, $n_{controls} = 553$), however, there was no evidence of mediation[38]. The lack of studies focused on high-dimensional mediation pathways in the relationships between toxicant mixtures and pregnancy outcomes signifies a major knowledge gap in environmental epidemiology research. Identification of high-dimensional mediation pathways helps prioritize biomonitoring efforts for early interventions. By further validating the mediation effect of the cytochrome p450 pathway in independent samples, we may potentially be able to utilize eicosanoid biomarkers of this pathway as early signals of disease states in pregnancy attribute to phthalate exposure.

There are features of our study that can be improved upon in order to advance knowledge of mediating biological pathways for pregnancy outcomes. Our study was not only limited in sample size, but also by a single cross-sectional assessment of exogenous and endogenous biomarkers. One previous study demonstrated significant changes in the expression of cytochrome p450 enzymes across pregnancy in placental tissue[39]. Another study has also shown that fetal expression levels of cytochrome p450 enzymes vary significantly from gestation to neonatal years[27]. Additionally, the cross-sectional design of our study also underlines the potential for reverse causation. For example, elevated inflammatory disease states from other causative agents may be linked to increased cytochrome p450 products, in turn impacting renal clearance of phthalate metabolites measured in urine samples. The cross-sectional nature of our study is also vulnerable to exposure measurement misclassification given that environmental toxicants such as phthalates, phenols, and polycyclic aromatic hydrocarbons have short half-lives in humans. However, when we previously investigated phthalates and preterm birth using four time points[15] and evaluated single time points cross-sectionally[40], we observed consistent associations between the repeated measures analysis and the time point that was used for this present study (26 weeks gestation). Single time point measurements of endogenous biomarkers further limit the inference of long-term biological effects. However, we can evaluate previous repeated measures studies of endogenous biomarkers and assess their intra-class correlation coefficients to quantify the reliability of single measurements. For example, a previous repeated measures study of select eicosanoids in women ($n = 9$) observed low intra-individual variability across eight repeated blood measurements[41]. In the LIFECODES cohort, we have also leveraged repeated measurements of the inflammatory biomarkers (CRP and cytokines), and observed moderate intra-individual variability[10]. These results provide some indication that although single time point measurements of eicosanoids and inflammatory markers are not ideal, they may have moderately good utility to assess biological effects. Future studies should build upon these findings to investigate environmental exposures and cytochrome p450 products as mediators in a longitudinal design in order to evaluate other vulnerable windows of development during pregnancy.

In regard to the methodological framework, multivariate mediator shrinkage, dimension reduction, and penalization were limited to a continuous outcome variable, and future innovations in statistical modeling will need to be developed in order to adapt our analytical framework for case-control settings with preterm birth. Another limitation of our study is the lack of potential confounders that may contribute to residual confounding. Specifically, dietary information may be an important confounder, given that some toxicants are associated with the type of food products that are consumed. To evaluate this, we conducted

sensitivity analyses to quantify the magnitude of association required by unmeasured confounders using two separate techniques (*medsens* function and E-value estimation).

Our study contains multiple strengths that underline its utility in epidemiologic research. We applied an extensive ensemble of statistical methods in our analytical framework. By doing so, we build upon the methodological advancements in high-dimensional mediation analysis by incorporating mixtures analysis—a critical and necessary pre-requisite for understanding the health consequences of the human exposome. We created a guided template for mediation analysis that is widely applicable in various settings and flexible to different structures of exposure data, mediator biomarkers, and health outcomes. Another strength of our study was the robust exposure assessment that quantified over four different classes of environmental agents that are known or suspected reproductive toxicants. Similarly, we measured a comprehensive panel of eicosanoids, which informs potential biomarkers for health studies.

In conclusion, this study integrates environmental mixtures analysis with cutting edge multivariate mediation approaches. Toxicants with sparse effects exhibited subtle mediation effects. However, cumulative exposures, assessed through the construction of the phthalate risk score, exhibited evidence for mediation through the cytochrome p450 pathway in the context of decreased gestational age at delivery. In order to accurately estimate the impact of the human exposome, future studies should consider the implementation of high-dimensional mediation analysis combined with exposure mixtures methods.

## Methods

**Study population.** The LIFECODES pregnancy cohort was developed at the Brigham and Women's Hospital in Boston, MA[40,42]. The overall cohort of 1600 pregnant women was recruited between 2006 and 2008. The Brigham and Women's Hospital administered institutional review board approval for this study. This study sample includes 161 women with complete exposure and endogenous biomarker measurements. Participants were over the age of 18 and recruited early in pregnancy (<15 weeks gestation). In total, this sample includes 52 cases of preterm birth (defined as delivery <37 weeks gestation), and 109 controls. Spontaneous preterm birth ($n_{cases} = 30$, $n_{controls} = 109$) was defined as either having a preterm premature rupture of the membranes, or spontaneous preterm labor[43]. Questionnaires and physical exams were administered at study visits to collect data on key covariates (e.g., maternal age, race, education, body mass index). Participants provided biological specimens (urine and blood) at a clinic visit occurring between 23.1 and 28.9 weeks gestation. Maternal urine samples were collected at a median of 26 weeks gestation and subsequently frozen at $-80\,°C$. Plasma was collected from 10 mL of blood draw using ethylenediaminetetraacetic acid plasma tubes. After collection, blood was stored at $+4\,°C$ for less than 4 h and subsequently centrifuged for 20 min before being stored at $-80\,°C$.

**Exposure analytes.** All exposure analytes were measured in urine samples by NSF International in Ann Arbor, MI. Quantification of individual toxicants and analytes followed protocols developed by the Centers for Disease Control and Prevention[44–47]. We used high-performance liquid chromatography-electrospray ionization tandem mass spectrometry (HPLC-ESI-MS/MS) to quantify a total of nine phthalate metabolites. We used isotope dilution liquid chromatography with tandem mass spectrometry (ID-LC–MS/MS) to quantify a total of eight polycyclic aromatic hydrocarbon metabolites and ten phenolic analytes. Finally, a total of 17 trace metals were measured using a Thermo Fisher (Waltham, MA, USA) iCAP RQ inductively coupled plasma mass spectrometer (ICPMS) with Teledyne CETAC Technologies (Omaha, NE, USA) ASX-520 autosampler.

**Endogenous biomarkers.** A panel of 51 eicosanoids and lipid metabolites were measured in plasma samples using a 6490 Triple Quadrupole mass spectrometer (Agilent, New Castle, DE, USA). The individual eicosanoids were identified using metabolite-specific fragmentation and retention times[12]. We used the Milliplex MAP High Sensitivity Human Cytokine Magnetic Bead Panel (EMD Millipore Corp., St. Charles, MO) to measure four cytokines: IL-1β, IL-6, TNF-α, and IL-10. An additional inflammation marker, CRP, was also measured using a DuoSet enzyme-linked immunosorbent assay (ELISA) (R&D Systems, Minneapolis, MN)[10].

We measured three protein oxidation markers in plasma samples: 3-nitrotyrosine [NY], 3-chlorotyrosine [CY], and *o,o'*-dityrosine [DY]. To quantify these biomarkers, total plasma protein was first precipitated and diluted with a

phosphate buffer. The samples were then delipidated, injected with isotopically labeled standards, and hydrolyzed for 24 h. Subsequently, the processed plasma samples underwent liquid chromatography-electrospray ionization tandem mass spectrometry. Additional oxidative stress markers 8-IP and 8-OHdG were measured in urine samples at Cayman Chemical (Ann Arbor, MI). 8-IP was quantified using affinity column chromatography followed by an enzyme immunoassay, and 8-OHdG was measured directly through an enzyme immunoassay[48].

**Statistical analyses**. All statistical analyses were conducted using R (version 3.5.1). Descriptive statistics were weighted based on calculated inverse probability weights that were corrected for over-representation of preterm birth cases in the sample compared to the larger LIFECODES cohort. The *survey* package (version 4.0) was used to account for inverse probability weights in all downstream analyses.

In Fig. 1, we illustrate a guide to the analytical framework that we developed for this multivariate mediation analysis that may be used in high-dimensional settings. Our mediation analysis builds on a counterfactual framework to formally define the causal effects. We can estimate natural direct and indirect effects from observed data with the following identifiability assumptions:

1. There is no unmeasured confounding for exposure-outcome relationship;
2. There is no unmeasured confounding for the mediator-outcome relationship after controlling for the exposure variable;
3. There is no unmeasured confounding for the exposure effect on all of the mediators;
4. There is no downstream effect of the exposure that confounds any mediator-outcome relationship

The above four assumptions are required to hold with respect to the whole set of mediators. Finally, as in all mediation analysis, the temporal ordering assumption also needs to be satisfied, i.e., the exposure precedes the mediators, and the mediators precede the outcome. Additional information on notations, definitions, and assumptions that we draw upon for our study are outlined in detail in supplemental materials, section 1.

**Approach 1a: Pairwise mediation of individual exposures and endogenous biomarkers**. The first approach in our analytical framework is a one-at-a-time pairwise mediation analysis with outcome $Y$ and every possible unique combination of analytes from the $p$-by-$n$ exposure matrix, $\mathbf{A}$ ($p = 38$, where $p$ is the number of exposure analytes, $n = 161$, where $n$ is the number of subjects), and endogenous biomarkers from the $q$-by-$n$ mediator matrix, $\mathbf{M}$ ($q = 61$, where $q$ is the number of mediators). This results in a total of 2,318 unique mediation models. The goal of this initial step is to conduct an exhaustive screen of all possible combinations of mediation pathways between each toxicant and endogenous biomarker measured in our study. The general framework for the mediation models is represented by:

$$M_i = \alpha_\alpha A_i + \boldsymbol{\alpha_c} \mathbf{C_i^T} + \epsilon_i \tag{1}$$

In these models, $\mathbf{C_i^T}$, is a covariate vector of $r$-by-*1* ($r = 7$, where the first element is scalar 1 for the intercept and the remainder elements are covariates) for the $i$-th subject. Each individual mediator is log-transformed and denoted one-at-a-time as $M_i$. The exposure variables are log-transformed and denoted one-at-a-time as $A_i$. In models where gestational age at delivery was the outcome variable, the conditional model for $\mathbf{Y}|\mathbf{A}, \mathbf{M}, \mathbf{C}$ is modeled as:

$$Y_i = \beta_a A_i + \beta_m M_i + \boldsymbol{\beta_c} \mathbf{C_i^T} + \delta_i \tag{2}$$

To account for the nested case-control sampling approach, we constructed inverse probability weights so that the subset sample was more representative of the proportion of cases and controls in the larger LIFCDOES cohort. Models in Eqs. (1) and (2) were weighted by these inverse probability weights. The natural indirect effect for each exposure-mediator pair is estimated by taking the product, $\alpha_a \times \beta_m$. When the conditional model for $\mathbf{Y}|\mathbf{A}, \mathbf{M}, \mathbf{C}$ modeled the binary outcome variables, such as preterm birth and spontaneous preterm birth, the model specification was:

$$\text{logit}\big[P\big(Y_i = 1|A_i, M_i, \mathbf{C_i^T}\big)\big] = \beta_a A_i + \beta_m M_i + \boldsymbol{\beta_c} \mathbf{C_i^T} + \delta_i \tag{3}$$

This first pairwise mediation analysis approach represents the baseline status quo in environmental epidemiology research, given that a majority of existing studies deploy a one-at-a-time assessment of mediation pathways. This approach may be most useful when there are few exposure variables with a targeted mediation hypothesis. However, it is susceptible to biased results when multiple exposure variables are highly correlated but are analyzed separately. The bias and hence inflated type 1 error result from possible confounding due to co-exposures missing from the analytic model. In addition, users can modify Approach 1a and utilize it as a screening step in multiple exposure settings. For example, mediation pairs that yield modest signals with relaxed significance thresholds estimated from Approach 1a may then be prioritized and passed onto Approach 2.

**Approach 1b: Mediation analysis with exposure-class risk scores**. In our analytical framework, we introduce an alternative approach for mediation analysis by proposing a method to construct environmental risk scores[23], thus reducing the dimensionality of the exposure matrix from 38 to 4, as well as reducing collinearity

across the exposure classes by using a summary risk score (Supplementary Table 4). The goal of this approach is to reduce bias due to collinearity and estimate the cumulative effect of toxicants based on their respective class, $k$ (phthalates [$p_1 = 9$], phenols [$p_2 = 9$], polycyclic aromatic hydrocarbons [$p_3 = 8$], and metals [$p_4 = 12$]). In this setting, we reduce the total number of pairwise mediation models to 244. We first estimate weights to construct environmental risk scores by applying adaptive elastic net regularization (*gcdnet* [version 1.0.5]) on toxicants based on their classes:

$$\widehat{\gamma_k} = \underset{\gamma_k}{\arg\min}\left\{\frac{1}{n}\sum_{i=1}^{n}\big(Y_i - \boldsymbol{\gamma_c}\mathbf{C_i^T} - \boldsymbol{\gamma_k}\mathbf{A_{ik}^T}\big)^2 + \lambda_1\sum_{j=1}^{p_k}|\gamma_{kj}| + \frac{\lambda_2}{2}\sum_{j=1}^{p_k}\gamma_{kj}^2\right\} \tag{4}$$

The adaptive elastic net combines the least absolute shrinkage and selection operator penalty with ridge regression[49]. In this setting, $Y_i$ is the continuous outcome variable gestational age at delivery. There are $p_k$ exposure analytes in the $k$-th exposure class, and $\mathbf{A_{ik}^T}$ is the $p_k$-by-1 column vector of log-transformed and standardized exposures for individual $i$($i = 1, \ldots, n$). In this application, $\lambda_1$ is a tuning parameter that performs shrinkage of coefficients for each exposure variable $j$($j = 1, \ldots, p_k$), and $\lambda_2$ operates as a tuning parameter used to stabilize solutions paths that navigate multi-collinearity in the exposure matrix. We utilized 5-fold cross-validations and optimization of prediction errors in order to estimate $\lambda_1$ and $\lambda_2$.

To construct environmental risk scores for each exposure class, we first extract $\boldsymbol{\gamma_k}$, which is a 1-by-$p_k$ vector of coefficients estimated by adaptive elastic net with its entries denoted as $\gamma_{kj}$, and designate them as weights. We then estimate each individual's scores for the $k$-th exposure class, the scalar $\text{ERS}_{ik}$, by calculating the linear combination using weights produced from the adaptive elastic net, applied across each individual's observed measurements of each toxicant:

$$\text{ERS}_{ik} = \widehat{\gamma_k}\mathbf{A_{ik}^T} \tag{5}$$

Next, we apply Eqs. (1) and (2) to estimate mediation signals. We input each class level $\text{ERS}_{ik}$ one-at-a-time as the single exposure variable, $A_i$. Since the $\text{ERS}_{ik}$ are standardized, mediators are also log-transformed and standardized in this setting.

Approach 1b overcomes issues of confounding due to multiple co-exposures and is less susceptible to bias from the correlation between exposure variables, chiefly because all exposures are included in a single multivariate model and the adaptive elastic net is robust to collinearity[49]. This approach may be most useful when users have several exposure variables and are interested in evaluating whole exposure classes to inform mixtures-based risk assessment rather than an individual toxicant analyte.

**Approach 2: Mediation analysis with multiple mediators**. The second approach of our analytical framework provides options for evaluating the $q \times n$ mediator matrix by either conducting Bayesian shrinkage estimation, reducing the dimensionality of the mediator matrix, or penalizing parameters in the mediation model. For shrinkage estimation, we applied Bayesian shrinkage estimation. In this setting, we deployed a high-dimensional shrinkage mediation analysis, one-at-a-time, for each exposure variable, $A_i$, as individual toxicants ($p = 38$) and in the dimensionally reduced exposure matrix using environmental risk scores for the $k$-th exposure class, $\text{ERS}_{ik}$ (Fig. 1).

**Option 2a: Bayesian shrinkage**. The major goal of Bayesian shrinkage estimation is to evaluate global mediation effects of multiple mediators simultaneously and extract posterior inclusion probability from a mixture prior as measures of importance among mediation pathways. For each exposure variable, $A_i$, we implemented shrinkage estimation by applying Bayesian mediation analysis with the *bama* package (version 0.9.1)[20]. In this application, *bama* modifies Eq. (1) into a multivariate regression model that jointly evaluates all mediators simultaneously in association with individual toxicants:

$$\mathbf{M_i^T} = \boldsymbol{\alpha_a} A_i + \boldsymbol{\alpha_c}\mathbf{C_i^T} + \boldsymbol{\epsilon_i} \tag{6}$$

$\mathbf{M_i^T}$ denotes the mediator column vector with a length equivalent to the number of mediators. There are two options for defining this vector: (1) using all mediators simultaneously, and (2) reviewing the literature to derive groups based on biological pathways or processes and then partitioning the mediator matrix into those groups. For the second option, we defined groups, $g$($g = 1, 2, 3, 4, 5$) corresponding to literature derived mediator groups (cyclooxygenase pathway [$q_1 = 14$], cytochrome p450 pathway [$q_2 = 18$], lipoxygenase pathway [$q_3 = 14$], parent lipid compounds [$q_4 = 5$], inflammation biomarkers [$q_5 = 5$]). Two mediator groups, oxidative stress biomarkers, and protein damage biomarkers each had less than 5 individual mediators, therefore we excluded them from this group-based analysis. Each of these groups were modeled one-at-a-time as $\mathbf{M_i^T}$ in Eq. (6). In this setting, $\boldsymbol{\alpha_a}$ is a $q_g \times 1$ vector of coefficients, and $\boldsymbol{\alpha_c}$ is a matrix of dimension $q_g \times r$, where $r$ is the number of confounders/covariates. Implementation of *bama* is restricted to continuous outcome variables; therefore, we focus on gestational age at delivery and modify the model in Eq. (2) to contain the entire mediator matrix:

$$Y_i = \beta_a A_i + \boldsymbol{\beta_m}\mathbf{M_i^T} + \boldsymbol{\beta_c}\mathbf{C_i^T} + \delta_i \tag{7}$$

Here $\beta_a$ is scalar, representing the estimated coefficient for the exposure given each

mediator group, $\mathbf{M_i^T}$, and $\boldsymbol{\beta_m}$ is a 1-by-$q_g$ vector of coefficients, where $q_g$ is equal to the number of total mediators or the number of mediators that belong in each group listed previously. In alignment with the continuous shrinkage method used in Bayesian sparse linear mixed models[50], *bama* assumes that the mediation effects are predominantly sparse, with selected active mediators having larger mediation effects. *Bama* utilizes the *L2* norm[51] to perform shrinkage and selection on the effects of $\boldsymbol{\alpha_a}$ and $\boldsymbol{\beta_m}$ separately, and for each exposure, $A_i$, estimates the natural indirect effect as the sum of the products of the individual effects $(\boldsymbol{\alpha_a})_\ell \times (\boldsymbol{\beta_m})_\ell$ and defines global pathway-specific mediation effects as:

$$\sum_{\ell=1}^{q_g} \left\{ (\boldsymbol{\alpha_a})_\ell (\boldsymbol{\beta_m})_\ell \right\}^2 \qquad (8)$$

where $(\boldsymbol{\alpha_a})_\ell$ and $(\boldsymbol{\beta_m})_\ell$ are the $\ell$-th entries of the vector $\boldsymbol{\alpha_a}$ and $\boldsymbol{\beta_m}$, respectively. Prior specification for $\boldsymbol{\alpha_a}$ and $\boldsymbol{\beta_m}$ is set to default settings in *bama*, which has been explained previously[20].

**Option 2b: Dimension reduction.** The secondary option in Approach 2 of the analytical framework is mediator dimension reduction. The main goal of reducing the dimensionality of the mediator matrix was to determine the extent to which individual toxicants are mediated by whole groups of endogenous biomarkers, either as the total mediator matrix of all biomarkers, or by biologically relevant groups (cyclooxygenase pathway, cytochrome p450 pathway, lipoxygenase pathway, parent lipid compounds, and inflammation biomarkers). Specifically, we constructed directions of mediation $\mathbf{w}$, a vector of the same size as the number of mediators considered, using population value decomposition (*PDM* [version 1.0])[19]. In this application, mediators are again organized into five groups, $g$. Each mediator within-group $g$ contributes to the group-specific $\mathbf{w_g}$(1-by-$q_g$) under the consideration of each exposure, $A_i$, thus when using the first direction of mediation for each group, we reduce the total dimension of the mediator matrix (from $q = 61$ individual mediators to $s = 5$ groups and also evaluate all mediators simultaneously). The direction of mediation, $\mathbf{w_g}$, is the coefficient vector of the linear combination of mediators in each group. Each $\mathbf{w_g}$ is estimated by maximizing the likelihood of structural equation modeling consisting of the models specified in Eqs. (6) and (7). The coefficients estimated in the first direction of mediation, $\mathbf{w_g}$, are then used as weights to estimate a mediator group effect, $\text{MGE}_{ig}$, by taking the linear combination of the coefficient weights across each measured mediator for each individual:

$$\text{MGE}_{ig} = \mathbf{w_g} \mathbf{M_{ig}^T} \qquad (9)$$

Here we use $\mathbf{M_{ig}^T}$ for the column vector of $q_g$ mediators in group $g$ for individual $i (i = 1, \ldots, n)$. We create an MGE for all mediators simultaneously or each mediator group (cyclooxygenase pathway, cytochrome p450 pathway, lipoxygenase pathway, parent lipid compounds, and inflammation biomarkers) as the new mediator variable. Then we conduct $(s + 1)$ pair-wise single mediation analyses for exposures (individually and as risk scores). We leverage Eqs. (1) and (2) and input each MGE as the single mediator, $M_i$. This mediation approach continues to focus strictly on gestational age at delivery, and we estimate the natural indirect effect by taking the simple product $\alpha_a \times \beta_m$. For the MGE that was constructed using all mediators simultaneously, we calculated correlation coefficients between the MGE and individual mediators to characterize which mediators contributed greatest toward the MGE.

Approach 2 overcomes issues surrounding the correlation between mediators, and combined with Approach 1b, analysis of the exposure matrix is also less biased by collinearity among exposure variables. This approach continues to utilize a mediation analysis with an overall single exposure variable being tested for mediation signals with a single mediator variable representative of whole mediator groups. As such, this approach is especially useful in multi-omics settings where biologically derived grouping structures are informed through the literature. Thus, inference can be more biologically drawn between exposure classes and biological pathways.

**Option 2c: Penalization.** Finally, using each ERS, we also explored two additional multivariate mediation methods: (1) pathway lasso[22], and (2) high-dimensional mediation analysis using a joint significance test for mediation effect[21]. These methods evaluate specific pairs of ERS and mediators to compare and differentiate group-level findings from sparse mediation effects of individual mediators. Both of these comparative methods deploy the models denoted in Eqs. (6) and (7).

Pathway lasso introduces a penalty parameter on the product of indirect effects and performs shrinkage and selection to identify pairs of ERS and single mediators that have the most robust mediation pathway.

We applied the joint significance test for high-dimensional mediation analysis using the *hima* package (version 1.0.7). *Hima* applies the minimax concave penalty (MCP) to regularize the coefficients estimated in Eqs. (6) and (7) and identify important mediating pathways[21]. Subsequently, *hima* conducts a joint significance test based on the MCP-penalized coefficients, resulting in adjusted p-values accounting for multiple comparisons using either the Benjamini Hochberg or Bonferroni method.

**Sensitivity analyses.** In addition to the dimension reduction approach with population value decomposition, we also explored a sparse principal components mediation analysis method[52]. In this approach, all mediators are evaluated simultaneously to construct individual mediator vectors similar to Eq. (9). Individual sparse principal components were modeled as mediators using Eqs. (1) and (2). We also evaluated correlations between sparse principal components and individual mediators to evaluate their contribution to the newly constructed components.

To evaluate the robustness of our proposed analytical approach toward the assumptions that we defined earlier, including sequential ignorability, we recommend deploying three additional sensitivity analyses. First, we leverage the *medsens* function within the *mediation* package (version 4.5.0). This sensitivity analysis simulates the change in the natural indirect effect as a function of a sensitivity parameter, $\rho$, which is representative of unmeasured confounders influencing the variation in the mediator and outcome models[53]. Another sensitivity analysis that we recommend is the calculation of an E-value to evaluate the vulnerability of adjusted mediation models to unmeasured confounding. To calculate this parameter, we used the *EValue* package (version 3.0.9) and focused only on our final model of phthalate risk score and the $\text{MGE}_i$ created from using all mediators simultaneously. In this setting, with a continuous outcome variable, we utilized the risk ratio approximation approach for standardized outcome variables[54]. A third sensitivity analysis that can be leveraged is the evaluation of the difference in mediation effect signal with the agnostic combination of all covariates in every unique combination. For six covariates in our study, that results in the evaluation of 63 unique covariate combinations.

## Data availability

The clinical outcomes and covariate data are not publicly available due to them confidentiality information that could compromise research participant privacy and confidentiality constraints through the existing Institutional Review Board. To address this, we generated a synthetic dataset with 161 observations, 6 covariates, and an outcome variable where the marginal distribution of the outcome mimics that of the relationship between the true outcome (gestational age at delivery) and the phthalate risk score in our actual dataset. The true measured and observed exposure and mediator biomarker data that support the findings of this study are also included in this dataset and provided in the cited github repository and can be readily used with the analytical framework.

## Code availability

Analytical framework in the form of R scripts available at https://github.com/umich-cphds/environmental_mediation_framework.

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

## Acknowledgments

Subject recruitment and sample collection were originally funded by Abbott Diagnostics. This work was also supported by the National Institute of Environmental Health Sciences, National Institutes of Health (grants R01ES018872, P42ES017198, P30ES017885, P50ES026049, and U2CES026553). Support for Kelly Ferguson was provided in part by the Intramural Research Program of the NIH, National Institute of Environmental Health Sciences (ZIAES103321).

## Author contributions

M.A. contributed to data pre-processing, data analysis, writing, and interpretation. Y.S. contributed to data analysis, writing, and interpretation. K.F. and D.C. contributed to data pre-processing and interpretation. L.Z. and S.P. contributed to eicosanoid measurement and quantification. J.M., T.M., and B.M. contributed to study design and interpretation.

## Competing interests

The authors declare they have no competing interests as defined by Nature Research that might be perceived to influence the results and/or discussion reported in this manuscript.
