## [Peer Review File · Nature Communications]

Reviewers' Comments:

Reviewer #1:

Remarks to the Author:

This manuscript introduces an analytical pipeline for mediation analysis with multiple exposures and multiple mediators. The paper aims to address an important question in mediation analysis. However, the assumptions required by the considered approaches may not hold. Therefore, the conclusions are questionable. The following are my detailed comments.

Major Concerns:

1. The number of mediators is $q = 63$. Depending on the sample size, I would be more careful calling it a high-dimensional mediation analysis. Especially, the sample size in the paper is $N = 161$, which is greater than the number of mediators.
2. In the study, multiple exposures, i.e., the $p = 38$ environmental toxicants, are considered. Are these exposures correlated? If so, the proposed mediation analysis, either the pair-wise or multi-mediator approaches, is biased. The mediation analysis approaches considered in the manuscript were designed with a single exposure. Application to a setting with multiple exposures is not straightforward.
3. A missing portion of the manuscript is the discussion of the causal assumptions. Mediation analysis is a causal concept. I could not imagine drawing causal conclusions without discussing/examining causal assumptions. The considered case in this study is more complex than a classic multiple-mediator setting, that multiple (possibly correlated) exposures are considered here. The authors should present the imposed assumptions for causal, discuss the validity of the assumptions in the dataset, propose potential solutions when the assumptions do not hold.
4. Section 2 Results.
 - (a) Figures 2-4. Details are missing in these figures. What is the y-axis in the figures? What's the color code? In Figure 2, why are there only three estimates with confidence intervals? Are these the only significant ones?
 - (b) Figure 3. The dashed line in the figure is the 0.05 threshold line. Should multiple testing be considered here? In addition, the pair-wise mediation analysis requires strong assumptions, i.e., independent exposures and causally independent mediators. Without discussing the validity of these assumptions, the pair-wise results are not trustworthy. See also my comments for Section 4.4 Step 1.
 - (c) Figure 4. Again, multiple testing correction should be considered here.
5. Section 4.3. Based on the description in this section, I only see 58 mediator biomarkers, i.e., 53 eicosanoids and lipid metabolites, 5 oxidative makers (3-NY, 3-CY, o,o'-DY, 8-IP and 8-OHdG). Can the authors explain what the rest 5 biomarkers are?
6. Section 4.4.
 - (a) Step 1. The assumption of models (1) and (2) is that all the exposures are mutually independent, and all the mediators are conditionally independent given the exposures. This assumption is not valid in the real data, then this screening step is fundamentally not correct. Based on the current application, I don't this assumption holds unless the authors can provide support from the literature.
 - (b) Step 1. My understanding of Step 1 is a screening step. However, it seems no exposure-mediator pair will be removed from the analysis in the following steps. Then, what is the purpose of Step 1? Not even mention that it is not a valid step.
 - (c) Step 2. Why is it a $q \times q$ mediator matrix?
 - (d) Step 2. Model (5). I do not see the advantage of conducting the mediation analysis for each group separately. The mediators could be correlated even for those who are not in the same group. If the Bayesian shrinkage approach can handle the high-dimensional setting, should having all the mediators in the same model be better? In addition, the sum of the number of mediators in each group does not match the total number of mediators, which is $\sum_{g=1}^7 q_g = 61 \neq 63$.
 - (e) Step 2. The dimension reduction approach. Again, it is really confusing why the analysis will be done for each group separately. For some of the groups, the number of mediators is really small, such as $q_6 = 2$. For this group, I don't see the necessity to do a dimension reduction. The goal of

considering a projection approach, either the direction of mediation (Ch'en et al., 2017) or the PCA-based approach (Huang and Pan, 2016) or a sparse PCA based approach (Zhao et al., 2020), is to create conditionally independent mediators (conditional on the exposure) so that it is valid to employ the marginal mediation analysis. Why not apply these approaches to all the mediators, and interpret each component? With either thresholding as in Ch'en et al. (2017) or regression-based sparsifying as in Zhao et al. (2020), each component is a group of biomarkers. One can examine if this grouping is consistent with the biological grouping.

(f) Step 2. My understanding of the Bayesian shrinkage approach and the dimension reduction approach are two parallel approaches. The users can choose one based on the study hypothesis. Based on the current description and the pipeline in Figure 1, both approaches should be performed consecutively. If this is the case, I do not see how the results from the Bayesian approach will be incorporated into the dimension reduction approach. The authors should clarify this.

(g) Step 3. If having Step 3, I don't see the reason for having Steps 2 and 3 separately. Why not combine Steps 2 and 3? Conduct a dimension reduction on the exposures first. With the reduced number of exposures, do Step 2.

(h) Step 3. Though in this step, exposures are grouped into clusters. It is still not guaranteed that these clusters are independent. The authors should consider adding discussions here.

7. The math notations are not consistent and are hard to follow. Please simplify and clarify math notations. For example, some parameters have both super- and sub-scripts, some superscripts are in parenthesis, some are not. When introducing new notations, it is better to add the dimension of the parameter. For example, M_{ig} is a q_g -dimensional vector, you can write $M_{ig} \in \mathbb{R}^{q_g}$.

Minors:

1. Eq. (12), should A_{im}^{top} be in A_{imj} ?

References

Ch'en, O. Y., Crainiceanu, C., Ogburn, E. L., Caffo, B. S., Wager, T. D., and Lindquist, M. A. (2017). High-dimensional multivariate mediation with application to neuroimaging data. *Biostatistics*, 19(2):121–136.

Huang, Y.-T. and Pan, W.-C. (2016). Hypothesis test of mediation effect in causal mediation model with high-dimensional continuous mediators. *Biometrics*, 72(2):402–413.

Zhao, Y., Lindquist, M. A., and Caffo, B. S. (2020). Sparse principal component based high-dimensional mediation analysis. *Computational Statistics & Data Analysis*, 142:106835.

Reviewer #2:

Remarks to the Author:

This well-written manuscript introduced an analytical pipeline for multivariate high-dimensional mediation analysis where both exposure and mediators are high-dimensional. This pipeline represents an ensemble of multiple state-of-the-art techniques and tools in variable selection, dimension reduction, and mediation testing. This proposed pipeline potentially has wide application scenarios to meet the new and emerging needs in the multi-omic analysis. As a showcase, the authors applied the pipeline to a subset of $n=161$ participants from a parent cohort (i.e., LIFECODES cohort) with $p=38$ environmental toxicants measured in urine as exposures, and $q=63$ endogenous biomarkers measured in plasma as mediators. The pipeline produced biologically meaningful results. Although the analysis may be underpowered given the limited sample size, and not strictly high-dimensional (p or $q \gg n$), these should not be considered as significant drawbacks- from bioinformatics and systematic biology perspective.

There are some comments to improve clarity:

1. The pairwise mediation step is disconnected in the pipeline, as the subsequential steps do not rely on the findings from the pairwise mediation test. The purpose of pairwise mediation analysis

was mainly for demonstrating the sparsity of signals (which is almost always the case in reality) and the superiority of the actual high-dimensional mediation analysis (steps 2 and 3). It would be better to make step 1 as a screening step with a relaxed p-threshold to be inclusive (or consider a similar technique like sure independent screening). Then step 2 will be based on the top mediation combinations for the subsequent dimensional reduction technique. In addition, this will also be beneficial when the pipeline is applied to the real high or ultra-high dimensional settings.

2. For a robust estimation of natural mediation effects, the sequential ignorability assumptions are strong and can be easily violated, in particular, given a large number of exposures and mediators involved. The authors could consider providing some strategies (e.g., the sensitivity test provided in the mediation package) and more details to discuss this important aspect.

3. (Minor) All subtitles in step 3 indicate both "exposure and mediator" dimensions reduction- this could be misleading. It should be just exposure dimensions reduction.

4. (Minor) The error term is missing in equation (3).

Reviewer #3:

Remarks to the Author:

The paper conducted by Aung et al., introduced an analytical pipeline for high-dimensional mediation analysis to identify mediation pathways in the relationship between environmental toxicants and gestational age at delivery. Though the study is well-conducted, and the results are interesting, I have several major concerns.

This study included 161 women (52 cases and 109 controls) from 1600 pregnancy women who attended The LIFECODES pregnancy cohort. It is unclear what method was used to select these women. Are these selected 161 women representatives of the overall cohort population? Did the authors match the demographic characters between included and excluded populations? Why only 161 women were selected? Please clarify if the statistical power is sufficient to generate precise estimations.

"Maternal urine samples were collected at median 26 weeks gestation". Why urine samples during other stages of pregnancy were not collected (or used)? Plenty of previous studies have shown high variability of urinary phthalate metabolites, metals, polycyclic aromatic hydrocarbon metabolites, and phenolic analytes among pregnant women. Unless the authors have ascertained that these exposure biomarkers are reliable during pregnancy, I don't believe a single measurement can tell the true association between exposure and adverse birth outcomes based on such a small sample size. Though the authors mentioned this point as a limitation of this study, more data is needed to support why this should or shouldn't be a concern.

I guess the blood sample was also collected at a single point in time from each woman. Similar to my concern raised above, did you check the reproducibility of 63 endogenous biomarkers? Did the authors ascertain that a single measurement is reliable to reflect the levels of these biomarkers over a period of days, weeks, or months? Why blood and urine samples collected at median 26 weeks gestation? Early and late pregnancies are also critical for fetal growth.

NCOMMS-20-23029
Response to reviewers

Title: Application of a novel analytical framework for multivariate mediation analysis of environmental data

Authors: Max T. Aung¹, Yanyi Song¹, Kelly K. Ferguson², David E. Cantonwine³, Lixia Zeng⁴, Thomas F. McElrath³, Subramaniam Pennathur^{4,5,6}, John D. Meeker⁷, Bhramar Mukherjee^{1,8}

Authors' Overall Remarks: We thank the managing editor for the kind invitation to revise our manuscript, and the reviewers for providing several important insights and suggestions. The revised manuscript has benefited tremendously from the changes that we've made based on the reviewers' comments. One major revision that we have incorporated based on reflection of all reviewers' comments that the editor should be aware of is the restructuring of our analytical pipeline. We renamed our contribution from "analytical pipeline" to "analytical framework" and thus changing individual "Steps" into "Approaches". We believe that this renaming will allow the readership to more clearly interpret our goals of recommending an ensemble of approaches to users, depending on the user's study goals and data constrictions. Therefore, readers may more clearly recognize that any one given "Approach" does not necessarily need to be implemented as a prerequisite for another "Approach" to be used and these are a set of parallel choices available to the user. A given approach achieves its own set of goals and interpretations. Additionally, as suggested by reviewer 1, we re-organized the exposure dimension reduction to be mentioned as *Approach 1b*, ahead of the mediator dimension reduction methods in *Approach 2*.

Reviewer #1 (Remarks to the Author):

This manuscript introduces an analytical pipeline for mediation analysis with multiple exposures and multiple mediators. The paper aims to address an important question in mediation analysis. However, the assumptions required by the considered approaches may not hold. Therefore, the conclusions are questionable. The following are my detailed comments.

Major Concerns:

1. The number of mediators is $q = 63$. Depending on the sample size, I would be more careful calling it a high-dimensional mediation analysis. Especially, the sample size in the paper is $N = 161$, which is greater than the number of mediators.

Authors' Response: We appreciate the reviewer's caution on the use of the term high-dimensional in the context of our pipeline. We have revised the title of the manuscript to "Application of a novel analytical framework for multivariate mediation analysis of environmental data". Throughout the manuscript, we've removed most instances of the term high-dimensional, except for discussing the fact that in principle the proposed framework may be used in high-dimensional settings in future studies.

2. In the study, multiple exposures, i.e., the $p = 38$ environmental toxicants, are considered. Are these exposures correlated? If so, the proposed mediation analysis, either the pair-wise or multi-mediator approaches, is biased. The mediation analysis approaches considered in the manuscript were designed with a single exposure. Application to a setting with multiple exposures is not straightforward.

Authors' Response: We thank the reviewer for presenting this opportunity to discuss the issue of correlation between environmental toxicants. First, we would emphasize that pair-wise mediation analysis is a one at a time single exposure approach that is implemented multiple times, and not a setting with multiple exposures simultaneously. Nonetheless, we recognize that correlated exposures, although analyzed separately, would yield biased results when evaluating all potential pair-wise mediation signals. This will be due to the possibility of confounding by co-exposures. Further, we agree with the reviewer that there are significant drawbacks associated with pair-wise mediation analysis, and our intention was to underscore that single one-at-a-time mediation analysis is largely the status quo in environmental epidemiology studies. Although any given epidemiologic study may only report a single exposure at a time, in aggregate, across multiple published studies, the overall body of scientific literature on this topic is demonstrative of the pair-wise mediation approach that we describe in *Approach 1a*. Accordingly, we recognized this opportunity to expand on our explanation of the issues surrounding *Approach 1a* as a baseline status quo method for users when they consider pair-wise mediation analysis as an analytical approach:

Section 4.4, paragraph 6:

“This first pairwise mediation analysis approach represents the baseline status quo in environmental epidemiology research, given that a majority of existing studies deploy a one-at-a-time assessment of mediation pathways. This approach may be most useful when there are few exposure variables with a targeted mediation hypothesis. However, it is susceptible to biased results when multiple exposure variables are highly correlated but are analyzed separately. The bias and hence inflated type 1 error results from possible confounding due to co-exposures missing from the analytic model. Additionally, users can modify Approach 1a and utilize it as a screening step in multiple exposure settings. For example, mediation pairs that yield modest signals with relaxed significance thresholds estimated from Approach 1a may then be prioritized and passed onto Approach 2.”

Additionally, we created and added a supplemental figure 1 (shown below) that illustrates the correlation matrix between the individual exposures. There were some moderate to high correlation coefficients ($\rho > 0.5$) within toxicant classes for phthalates, polycyclic aromatic hydrocarbons, and trace metals. The highest correlation documented was $\rho = 0.97$ between the phthalate metabolites MEOHP and MEHHP. However, we did not observe substantial correlation across toxicant classes. This situation is therefore ameliorated by our exposure dimension reduction approach by constructing risk scores specific to toxicant classes. We provided an additional supplemental table 5 (shown below) reporting correlation coefficients between environmental risk scores, indicating weak correlation between the risk scores. The correlation between risk scores were between |0.03| and |0.26|. We have added additional text about these considerations:

In section 4.4, paragraph 11:

“Approach 1b overcomes issues of confounding due to multiple co-exposures and is less susceptible to bias from correlation between exposure variables, chiefly because all exposures are included in a single multivariate model and adaptive elastic net is robust to collinearity⁴⁸. This approach may be most useful when users have several exposure variables and are interested in evaluating whole toxicant classes to inform mixtures-based risk assessment rather than an individual toxicant analyte.”

In section 2.4, paragraph 1:

*“Prior to exposure dimension reduction, individual toxicants had high within-class correlation, with the highest correlation coefficient ($\rho=0.97$) observed between the phthalate metabolites MEOHP and MEHHP (**Supplemental Figure 2**). Creating environmental risk scores removed within class*

correlations and we observed low correlation coefficients between risk scores (ρ ranging from $|0.03 - 0.26|$) (Supplemental Table 4).”

Supplemental Figure 1. Correlation matrix of individual environmental toxicants. Estimates are based on specific gravity adjusted concentrations and weighted for inverse probability weights.

Supplemental Table 4. Correlation Matrix Between Toxicant Risk Scores

	Phthalate Risk Score	Phenol Risk Score	PAH Risk Score	Metal Risk Score
Phthalate Risk Score	1	-0.15	-0.23	0.03
Phenol Risk Score	-0.15	1	0.26	-0.04
PAH Risk Score	-0.23	0.26	1	0.04
Metal Risk Score	0.03	-0.04	0.04	1

3. A missing portion of the manuscript is the discussion of the causal assumptions. Mediation analysis is a causal concept. I could not imagine drawing causal conclusions without discussing/examining causal assumptions. The considered case in this study is more complex than a classic multiple-mediator setting, that multiple (possibly correlated) exposures are considered here. The authors should present the imposed assumptions for causal, discuss the validity of the assumptions in the dataset, propose potential solutions when the assumptions do not hold.

Authors' Response: We appreciate the reviewer's comment regarding the causal assumptions and apologize for missing these important details in the previous version of our manuscript. We have a methodological paper that describes the technical assumptions underlying this setting in granular mathematical detail¹. We explain the assumptions here in more pragmatic language and refer the reader to the technical paper as well as present a brief mathematical summary in the supplementary material,

With multiple exposures, in *Approach 1b* we attempt to reduce the dimensionality of the exposures by an initial step of constructing environmental risk scores corresponding to each toxicant class. We then treat each of the risk scores as a single exposure variable and perform mediation analysis with multiple mediators and one exposure at a time. Therefore, the mediation models considered in this manuscript essentially involve one exposure, multiple mediators and the outcome of interest. We mention having mediation methods with multiple exposures and mediators as a potential extension in the discussion.

We have described an overview of the assumptions in section 4.4, paragraph 2:

“Our mediation analysis builds on a counterfactual framework to formally define the causal effects. We can estimate natural direct and indirect effects from observed data with the following identifiability assumptions:

1. *There is no unmeasured confounding for exposure-outcome relationship;*
2. *There is no unmeasured confounding for the mediator-outcome relationship after controlling for the exposure variable;*
3. *There is not unmeasured confounding for the exposure effect on all of the mediators;*
4. *There is no downstream effect of the exposure that confounds any mediator-outcome relationship*

The above four assumptions are required to hold with respect to the whole set of mediators. Finally, as in all mediation analysis, the temporal ordering assumption also needs to be satisfied, i.e., the exposure precedes the mediators, and the mediators precede the outcome. Additional information on notations, definitions, and assumptions that we draw upon for our study are outlined in detail in supplemental materials, section 1.”

And we have also provided the below text into the supplemental material:

Section 1. Notations, definitions, and assumptions.

Suppose our analysis is based on a study of n subjects, and for each subject $i, i = 1, \dots, n$, we collect data on the exposure A_i , q candidate mediators $\mathbf{M}_i = (M_i^{(1)}, M_i^{(2)}, \dots, M_i^{(q)})$, the outcome Y_i , r covariates $\mathbf{C}_i = (C_i^{(1)}, C_i^{(2)}, \dots, C_i^{(r)})$. To formally define causal effects and draw causal conclusions, we adopt a counterfactual framework for causal mediation analysis in the presence of multiple mediators. We define $\mathbf{M}_i(a) = (M_i^{(1)}(a), M_i^{(2)}(a), \dots, M_i^{(q)}(a))$ as the i -th subject's counterfactual value of the q mediators if he/she received exposure a , and define $Y_i(a, \mathbf{m})$ as this subject's counterfactual outcome under exposure level at a and mediators at $\mathbf{m} = (m^{(1)}, m^{(2)}, \dots, m^{(q)})$. With these notations, we can formally define the direct effect and effect mediated through the multiple mediators, i.e. indirect effect. The natural direct effect (NDE) is defined as $Y_i(a, \mathbf{M}_i(a^*)) - Y_i(a^*, \mathbf{M}_i(a^*))$, which is the change in the counterfactual outcomes when exposure changes from a^* (the reference level) to a while hypothetically controlling mediators at the level that they would have naturally been with exposure a^* . The natural indirect effect (NIE) is defined as $Y_i(a, \mathbf{M}_i(a)) - Y_i(a, \mathbf{M}_i(a^*))$, the change in counterfactual outcomes when mediators change from $\mathbf{M}_i(a^*)$ to $\mathbf{M}_i(a)$ while fixing the exposure at a . The total effect (TE) can

then be expressed as the summation of the NDE and the NIE: $Y_i(a, \mathbf{M}_i(a)) - Y_i(a^*, \mathbf{M}_i(a^*)) = Y_i(a, \mathbf{M}_i(a)) - Y_i(a, \mathbf{M}_i(a^*)) + Y_i(a, \mathbf{M}_i(a^*)) - Y_i(a^*, \mathbf{M}_i(a^*)) = \text{NDE} + \text{NIE}$.

The counterfactual variables used to define causal effects are not necessarily observed, but the identification of causal effects must be based on observed data. Therefore, further assumptions regarding the confounders are needed for the identification and interpretation of causal effects². We will use $A \perp B \mid C$ to denote that A is independent of B conditional on C . To estimate the average NDE and NIE from observed data, the following identifiability assumptions are required:

- (1) $Y_i(a, \mathbf{m}) \perp A_i \mid \mathbf{C}_i$, that is, no unmeasured confounding for exposure-outcome relationship;
- (2) $Y_i(a, \mathbf{m}) \perp \mathbf{M}_i \mid \{\mathbf{C}_i, A_i\}$, that is, no unmeasured confounding for any of mediator-outcome relationship after controlling for the exposure;
- (3) $\mathbf{M}_i(a) \perp A_i \mid \mathbf{C}_i$, that is, no unmeasured confounding for the exposure effect on all the mediators;
- (4) $Y_i(a, \mathbf{m}) \perp \mathbf{M}_i(a^*) \mid \mathbf{C}_i$, that is, no downstream effect of the exposure that confounds any mediator-outcome relationship.

The above four assumptions are required to hold with respect to the whole set of mediators. Finally, as in all mediation analysis, the temporal ordering assumption also needs to be satisfied, i.e., the exposure precedes the mediators, and the mediators precede the outcome.

First, we note that the identifiability assumptions cannot be verified empirically from the observed data³, and we can only justify our selection of covariates based on scientific knowledge. For both the outcome and mediator model, we adjust for specific gravity, maternal age, race, BMI at initial study visit, education level and health insurance provider as confounders. For assumption (1), we believe that those available covariates are natural to control for exposure-outcome confounders based on existing domain-specific literature⁴. For assumption (2), within each exposure level, those covariates are also natural to control for confounders that are associated with both endogenous biomarkers and gestational age at delivery⁵. For assumption (3), we included all the important potential exposure-mediator confounders as in another similar study⁶. Assumption (4) is usually a challenging condition to justify as it simultaneously involves counterfactuals with $A_i = a$ and $A_i = a^*$, one of which will not be observed in real data. The influence of violating the above identifiability assumptions can be assessed using sensitivity analysis, which has been well-developed for the single mediator setting⁷, and additional work is required to extend that approach to the multiple-mediator setting.

Regarding the temporal assumptions, in the LIFECODES birth cohort, participants provided biological specimens (urine and blood) at a clinic visit occurring between 23.1 and 28.9 weeks gestation. All exposure analytes were measured in urine samples, and the potential mediators, a large panel of 53 eicosanoids and lipid metabolites, were measured in plasma samples. The outcome of interest, gestational age of the newborn was recorded at delivery. Therefore, the exposure and mediators come earlier than the outcome variable. While it is hard to disentangle the temporal ordering between the exposure and mediators' measurements, our conceptual model supports the statistical model. Exposure to toxicants (e.g. phthalates, toxic heavy metals) may disturb receptor activity and induce their responses, which could affect the signaling molecules related to inflammation and metabolism. Biomarkers of inflammation and oxidative stress have been shown to be associated with the risk of preterm birth.

References:

1. Song, Y. *et al.* Bayesian shrinkage estimation of high dimensional causal mediation effects in omics studies. *Biom biom.*13189–11 (2019). doi:10.1111/biom.13189

2. Vanderweele, T. J. & Vansteelandt, S. Mediation Analysis with Multiple Mediators. *Epidemiologic Methods* **2**, 95–115 (2014).
3. Rubin, D. B. & Little, R. *Statistical Analysis with Missing Data, Vol. 793*. (2019).
4. Ferguson, K. K., McElrath, T. F., Ko, Y.-A., Mukherjee, B. & Meeker, J. D. Variability in urinary phthalate metabolite levels across pregnancy and sensitive windows of exposure for the risk of preterm birth. *Environment International* **70**, 118–124 (2014).
5. Aung, M. T. *et al.* Prediction and associations of preterm birth and its subtypes with eicosanoid enzymatic pathways and inflammatory markers. *Scientific Reports* 1–17 (2019). doi:10.1038/s41598-019-53448-z
6. Ferguson, K. K. *et al.* Mediation of the Relationship between Maternal Phthalate Exposure and Preterm Birth by Oxidative Stress with Repeated Measurements across Pregnancy. *Environ Health Perspect* **125**, 488–494 (2017).
7. Imai, K., Keele, L. & Yamamoto, T. Identification, Inference and Sensitivity Analysis for Causal Mediation Effects. *Statist. Sci.* **25**, 51–71 (2010).

4. Section 2 Results.

(a) Figures 2-4. Details are missing in these figures. What is the y-axis in the figures? What's the color code? In Figure 2, why are there only three estimates with confidence intervals? Are these the only significant ones?

Authors' Response: We appreciate the reviewer's request for clarification on figure details to enhance the manuscript's interpretability for the readership. There are technically no y-axes in figures 2-4 in terms of quantified parameters or estimates, but rather the vertical space is a placeholder for each predictor variable. The color coding is to provide quick visual delineation between toxicant classes. Both of these topics have clarified this in the figure captions stating:

"Toxicants are organized on the y-axis and color coded to differentiate classes: orange (phthalates), purple (phenols and parabens), green (polycyclic aromatic hydrocarbons), and grey (trace metals)."

The reviewer's question about why there are only three estimates with confidence intervals raises a concern. Figure 2 should have point estimates and confidence intervals for every single toxicant, not only three estimates with confidence intervals. It is possible that uploading the figure into a word document through the Nature Communications upload system altered figure components, as this has happened previously with other journal submission systems. We will reach out to the Editor about one potential solution for this issue, which is to upload the entire document as a PDF instead of word document.

(b) Figure 3. The dashed line in the figure is the 0.05 threshold line. Should multiple testing be considered here? In addition, the pair-wise mediation analysis requires strong assumptions, i.e., independent exposures and causally independent mediators. Without discussing the validity of these assumptions, the pair-wise results are not trustworthy. See also my comments for Section 4.4 Step 1.

(c) Figure 4. Again, multiple testing correction should be considered here.

Authors' Response: We appreciate the reviewer's suggestion to provide information on multiple testing. We adjusted for false-discovery and as suspected with the very small sample size, the observations were not robust to false-discovery. We have added a note about this into the figure caption stating that q-values for point estimates exceeded 0.2.

5. Section 4.3. Based on the description in this section, I only see 58 mediator biomarkers, i.e., 53 eicosanoids and lipid metabolites, 5 oxidative makers (3-NY, 3-CY, o,o'-DY, 8-IP and 8-OHdG). Can the authors explain what the rest 5 biomarkers are?

Authors' Response: The 5 additional biomarkers not mentioned in the above comment are 4 cytokines (IL-1 β , IL-6, TNF- α , and IL-10) and the inflammation marker C-reactive protein. Additionally, we corrected a typo that should read 51 eicosanoids, not 53.

6. Section 4.4.

(a) Step 1. The assumption of models (1) and (2) is that all the exposures are mutually independent, and all the mediators are conditionally independent given the exposures. This assumption is not valid in the real data, then this screening step is fundamentally not correct. Based on the current application, I don't this assumption holds unless the authors can provide support from the literature.

Authors' Response: We thank the reviewer for guiding the discussion on this important topic. We agree with the reviewer that an exhaustive step-wise approach has inherent biases due to correlation structures between exposures and mediators. After reflecting on the reviewer's comments, we more clearly contextualized our *Approach 1a* of pairwise mediation analysis as the baseline "status-quo" in health research. Several epidemiologic studies focus on one chemical analysis at a time. When evaluating the literature in aggregate across multiple published studies from the same observational cohorts, each focusing on a single compound, there are obvious issues of independence in exposures as the reviewer mentioned. We state this bias in our methods section as a disadvantage of pairwise mediation analysis and propose Approach 1b as an alternative and potential solution to reduce the collinearity issue:

Section 4.4 paragraph 6:

"This first pairwise mediation analysis approach represents the baseline status quo in environmental epidemiology research, given that a majority of existing studies deploy a one-at-a-time assessment of mediation pathways. This approach may be most useful when there are few exposure variables with a targeted mediation hypothesis. However, it is susceptible to biased results when multiple exposure variables are highly correlated but are analyzed separately. The bias and hence inflated type I error results from possible confounding due to co-exposures missing from the analytic model."

Section 4.4 paragraph 7:

"In our analytical framework we introduce an alternative approach for mediation analysis by proposing a method to construct environmental risk scores²³, thus reducing the dimensionality of the exposure matrix of dimension from 38 to 4 as well as reducing collinearity across the toxicant classes by using a summary risk score (Supplementary Table 4). The goal of this approach is to reduce bias due to collinearity and estimate the cumulative effect of toxicants based on their respective class, m (phthalates [$p_1 = 9$], phenols [$p_2 = 9$], polycyclic aromatic hydrocarbons [$p_3 = 8$], and metals [$p_4 = 12$])."

(b) Step 1. My understanding of Step 1 is a screening step. However, it seems no exposure-mediator pair will be removed from the analysis in the following steps. Then, what is the purpose of Step 1? Not even mention that it is not a valid step.

Authors' Response: We thank the reviewer's inquiry about the rationale for the pair-wise mediation approach of the proposed analytical pipeline (now named analytical framework). We would like to

clarify that the pairwise mediation analysis is not an initial screening step for the multivariate mediation analysis model. The goal of presenting this approach was to evaluate whether mediation signals could be detected at the individual toxicant and mediator biomarker level and by doing so assess the sparsity of signals. Additionally, the pairwise mediation analysis represents the status quo that exists in many studies of mediation analyses, where researchers often test one-at-a-time combinations of biomarkers for exposures and mediators. Another utility of pairwise mediation is that currently many environmental policies and regulations focus on legislation and risk assessments for single toxicants. Therefore, it is sometimes more directly applicable and necessary for researchers to conduct pairwise mediation analysis and provide specific mediation signals for a given exposure analyte.

(c) Step 2. Why is it a $q \times q$ mediator matrix?

Authors' Response: We have modified the text and a critical typo and now denoted the mediator matrix to be a $q \times n$ matrix.

(d) Step 2. Model (5). I do not see the advantage of conducting the mediation analysis for each group separately. The mediators could be correlated even for those who are not in the same group. If the Bayesian shrinkage approach can handle the high-dimensional setting, should having all the mediators in the same model be better? In addition, the sum of the number of mediators in each group does not match the total number of mediators, which is $\sum_{g=1}^7 q_g = 61 \neq 63$.

Authors' Response: We thank the reviewer for the opportunity to justify the grouping approach of our analytical framework. When developing the analytical framework, our goal was to approach the mediation analysis with biologically focused hypothesis testing that could be widely applicable in multi-omics settings. Several omics studies have the capacity to cluster mediator biomarkers by literature derived biological pathways or groups. This allows for targeted understanding of mediation pathways for specific environmental exposures. As the reviewer recommended, we also reported results where we did perform dimension reduction on all mediators simultaneously, and these results are reported in Figure 5 in the first row with the label "All Biomarkers" (see below and All Biomarkers is additionally emphasized with a red circle). Therefore, we created a mediator group effect (MGE) using "All Biomarkers", in addition to creating MGE for each biological group derived through the literature. We have added additional text to describe the approach of analyzing all mediators simultaneously:

Section 4.4, paragraph 13:

" \mathbf{M}_i^T denotes the mediator column vector with a length equal to the number of mediators. There are two options for defining this vector: (1) using all mediators simultaneously, and (2) reviewing the literature to derive groups based on biological pathways or processes and then partitioning the mediator matrix into groups."

Section 4.4, paragraph 15:

"The secondary option in Approach 2 of the analytical framework is mediator dimension reduction. The main goal of reducing the dimensionality of the mediator matrix was to determine the extent to which individual toxicants are mediated by whole groups of endogenous biomarkers, either as the total mediator matrix of all biomarkers, or by biologically relevant groups (cyclooxygenase pathway, cytochrome p450 pathway, lipoxigenase pathway, parent lipid compounds, and inflammation biomarkers)."

Section 4.4, paragraph 16:

“We create an MGE for all mediators simultaneously or each mediator group (cyclooxygenase pathway, cytochrome p450 pathway, lipoxygenase pathway, parent lipid compounds, and inflammation biomarkers,) as the new mediator variable, and conduct pair-wise single mediation. Then we conduct (s +1) pair-wise single mediation analyses for exposures (individually and as risk scores. We leverage equations 1 and 2 and input each MGE as the single mediator, M_i . This mediation approach continues to focus strictly on gestational age at delivery, and we estimate the natural indirect effect by taking the simple product $\alpha_a \times \beta_m$. For the MGE that was constructed using all mediators simultaneously, we calculated correlation coefficients between the MGE and individual mediators to characterize which mediators contributed greatest towards the MGE.”

Section 4.4, paragraph 17:

“Approach 2 overcomes issues surrounding correlation between mediators, and combined with Approach 1b, analysis of the exposure matrix is also less affected by the potential collinearity among exposure variables. This approach continues to utilize a mediation analysis with an overall single exposure variable (the summary risk score) being tested for mediation signals with the pre-defined mediator groups. As such, this approach is especially useful in multi-omics settings where biologically derived grouping structures are informed through the literature. Thus, inference can be more biologically drawn between toxicant classes and biological pathways.”

Figure 5. Forest plot of mediation analysis for environmental risk scores, mediator group effects, and gestational age at delivery. Models adjusted for specific gravity, maternal age, race, and BMI at initial study visit. Estimates have corresponding q-values exceeding 0.2. Abbreviations: Natural direct effect (NDE); Natural indirect effect (NIE)

For the issue about the sum of mediators, we have corrected a typo stating that the total number of mediators should be 61, not 63. This correction is also noted in the analytical framework diagram Figure 1:

Figure 1. Analytical framework for conducting high-dimensional mediation analysis of toxicant mixtures, endogenous biomarkers, and birth outcomes.

(e) Step 2. The dimension reduction approach. Again, it is really confusing why the analysis will be done for each group separately. For some of the groups, the number of mediators is really small, such as $q_6 = 2$. For this group, I don't see the necessity to do a dimension reduction. The goal of considering a projection approach, either the direction of mediation (Chén et al., 2017) or the PCA-based approach (Huang and Pan, 2016) or a sparse PCA based approach (Zhao et al., 2020), is to create conditionally independent mediators (conditional on the exposure) so that it is valid to employ the marginal mediation analysis. Why not apply these approaches to all the mediators, and interpret each component? With either thresholding as in Chén et al. (2017) or regression-based sparsifying as in Zhao et al. (2020), each component is a group of biomarkers. One can examine if this grouping is consistent with the biological grouping.

Authors' Response: We recognize and agree with the reviewer that groups with very small number of mediators is not ideal, nor is it particularly necessary to conduct dimension reduction in those settings. We have modified our methods section to adapt to this issue:

Section 4.4, paragraph 13:

“Let \mathbf{M}_i^T denotes the mediator vector with a length equivalent to the number of mediators. There are two options for to defining this vector: (1) using all mediators simultaneously, and (2) reviewing the literature to derive groups based on biological pathways or processes and then partitioning the mediator matrix into groups. For the second option, we defined groups, g ($g = 1,2,3,4,5$) corresponding to literature derived mediator groups (cyclooxygenase pathway [$q_1 = 14$], cytochrome p450 pathway [$q_2 = 18$], lipoxygenase pathway [$q_3 = 14$], parent lipid compounds [$q_4 = 5$], inflammation biomarkers [$q_5 = 5$]). Two mediator groups, oxidative stress biomarkers, and protein damage biomarkers each had less than 5 individual mediators, therefore we excluded them form this group-based analysis. Each of these groups were modeled one-at-a-time as \mathbf{M}_i^T in equation 6.”

In regard to the reviewer’s question about evaluating all mediators simultaneously and then applying marginal mediation analysis and interpreting components: our original rationale for clustering based on groups was intended on allowing users to leverage biologically driven hypothesis testing based on the literature on biomarkers or multi-omics assays. By doing so, researchers can characterize mediation signals that are most relevant for biological pathways and thus therapeutic or precision health approaches for early prediction and interventions. We do recognize the methodological advantage of additionally deploying a data driven approach in the absence of biological derived grouping structures. Therefore, we have added two additional data exploration approaches: (1) evaluating the correlation between the population value decomposition derived mediator group effect, MGE , and the individual mediators; and (2) sensitivity analysis based on the sparse principal components mediation analysis method recommended by the reviewer. We learned through both approaches that cytochrome p450 pathway mediators were much more influential in dimensionally reduced and constructed variables, which is consistent with our observation that this pathway yielded a mediation signal much more pronounced than other mediator groups when doing group-based MGE construction. Results for both approaches have been added to supplemental results (shown below for reference) and discussed in the methods and results sections:

Section 4.4, paragraph 16:

“For the MGE that was constructed using all mediators simultaneously, we calculated correlation coefficients between the MGE and individual mediators to characterize which mediators contributed greatest towards the MGE .”

Section 4.4, paragraph 21:

“As a sensitivity analysis, in addition to the dimension reduction approach that we leveraged, we also explored a sparse principal components mediation analysis method introduced by Zhao and colleagues⁵¹. In this approach, all mediators are evaluated simultaneously to construct individual mediator vectors similar to equation 9. Individual sparse principal components were modeled as mediators using equations 1 and 2. We also evaluated correlations between sparse principal components and individual mediators to evaluate their contribution to the newly constructed components.”

Section 2.3, paragraph 1:

*“To investigate the contributions of individual mediators towards the direction of mediation constructed for all mediator simultaneously, we illustrated correlation coefficients between the direction of mediation and individual mediators in **Supplemental Figure 1**. Only two mediators, RVD1 (lipoxygenase pathway) and 11,12-DHET (cytochrome p450 pathway) exhibited Pearson correlation coefficients greater than |0.3| with the direction of mediation constructed for all mediators simultaneously, indicating that these two pathways may have larger influence in this dimensionally reduced mediator vector.”*

Section 2.5, paragraph 3:

*“An alternative approach to population value decomposition derived mediator group effects is the use of sparse principal component constructed single mediators. This approach assumes sparse effects on the outcome variable among individual mediators. Sparse principal component mediation analysis yielded 24 principal components that accounted for 80% of the variance in the mediator matrix. There was no single sparse principal component that exhibited a significant mediation signal between the phthalate risk score and gestational age at delivery, however, a majority (66%) of the components had natural indirect effect estimates in the same direction as what we observed when evaluating all biomarkers simultaneously using population value decomposition (**Supplemental Table 7**). When evaluating contributions in loading factors for each sparse principal component, we observed that mediators from the cytochrome p450 pathway had the greatest number of loadings above 0.3 compared to other mediator groups (**Supplemental Figure 3**).”*

Supplemental Figure 1. Bar chart of correlation coefficients between the first direction of mediation for phthalate risk score and individual mediators. Estimates are weighted for inverse probability weights. Orange bars represent correlation coefficients exceeding |0.3|.

Supplemental Figure 3. Heat map of loading values for individual mediators relative to individual principal components estimated from sparse principal component based mediation analysis. Red and blue grids indicate positive and negative loadings, respectively. Color intensities represent the magnitude of loading, i.e. darker grids indicate greater loading. Mediators with loadings exceeding $|0.3|$ are labeled by green squares.

(f) Step 2. My understanding of the Bayesian shrinkage approach and the dimension reduction approach are two parallel approaches. The users can choose one based on the study hypothesis. Based on the current description and the pipeline in Figure 1, both approaches should be performed consecutively. If this is the case, I do not see how the results from the Bayesian approach will be incorporated into the dimension reduction approach. The authors should clarify this.

Authors' Response: We appreciate the reviewer's emphasis on clarification for our illustration of methods in step 2 (now *Approach 2*). We have clarified in the figure (shown below for reference) to indicate that users can utilize the two methods independently, and that Bayesian shrinkage does not feed directly into dimension reduction using population value decomposition or vice versa. In the methods section we clarified this further:

Section 4.4, paragraph 12:

“The second approach of our analytical framework provides options for evaluating the $q \times n$ mediator matrix by either conducting shrinkage estimation, reducing the dimensionality of the mediator matrix, or penalizing parameters in the mediation model.”

Figure 1. Analytical framework for conducting high-dimensional mediation analysis of toxicant mixtures, endogenous biomarkers, and birth outcomes.

(g) Step 3. If having Step 3, I don't see the reason for having Steps 2 and 3 separately. Why not combine Steps 2 and 3? Conduct a dimension reduction on the exposures first. With the reduced number of exposures, do Step 2.

Authors' Response: We thank the reviewer for presenting this question about differentiating the utility of having steps 2 and 3 separately. Based on the reviewer's recommendation, we modified Figure 1 and our methods section to discuss exposure dimension reduction first. However, we still continue to provide results showing individual toxicants and mediator dimension reduction in the results section. There are two main reasons for showing these results and considering it as a valid option. The first reason is grounded in historical approaches for policy recommendations. Many policy makers are limited to implementing regulations or inform risk assessments on individual toxicants, therefore, to be thorough

and provide research findings for specific toxicants was necessary in providing toxicant level risk profiling through mediation analysis. The second reason is for flexibility of our analytical framework for users. In many environmental epidemiology studies, some investigators may have exposure assessment of several toxicants (making *Approach 1b* more interesting and robust), but some investigators may be limited to having exposure assessment of only a few toxicants. In the situation with limited exposure assessment, *Approach 1a* combined with reduced mediator matrix would be more relevant and applicable for users.

(h) Step 3. Though in this step, exposures are grouped into clusters. It is still not guaranteed that these clusters are independent. The authors should consider adding discussions here.

Authors' Response: We appreciate the reviewer's emphasis on discussing concerns of co-dependency among exposure clusters. We have provided additional details on the correlation between environmental risk scores in supplemental table 5, also shown below. In this particular setting, the Pearson correlation coefficients between environmental risk scores were moderately low, ranging from |0.03 – 0.26|.

Supplemental Table 5. Correlation Matrix Between Toxicant Risk Scores

	Phthalate Risk Score	Phenol Risk Score	PAH Risk Score	Metal Risk Score
Phthalate Risk Score	1	-0.15	-0.23	0.03
Phenol Risk Score	-0.15	1	0.26	-0.04
PAH Risk Score	-0.23	0.26	1	0.04
Metal Risk Score	0.03	-0.04	0.04	1

7. The math notations are not consistent and are hard to follow. Please simplify and clarify math notations. For example, some parameters have both super- and sub-scripts, some superscripts are in parenthesis, some are not. When introducing new notations, it is better to add the dimension of the parameter. For example, M_{ig} is a q_g -dimensional vector, you can write $M_{ig} \in \mathbb{R}^{q_g}$.

Authors' Response: We are thankful that the reviewer provided this constructive critique about our math notations. We have revised our equations, and significantly simplified the math notations to reduce redundancies and make notations more uniform and consistent including use of suffixes and indices.

Minors:

1. Eq. (12), should A_{im}^{\top} be in A_{imj} ?

Authors' Response: We have modified this equation (now equation 5) to:

$$(5) ERS_{ik} = \widehat{\gamma}_k A_{ik}^T$$

And clarified that γ_k is a 1-by- p_k vector of coefficients estimated by adaptive elastic net and A_{ik}^T represents a p_k -by-1 column vector of all exposures in a given class, k , and not just one single exposure. Thus ERS_{ik} is a scalar for subject i corresponding to their exposure summary in class k .

References:

1. Chén, O. Y., Crainiceanu, C., Ogburn, E. L., Caffo, B. S., Wager, T. D., and Lindquist, M. A. (2017). High-dimensional multivariate mediation with application to neuroimaging data. *Biostatistics*, 19(2):121–136.
2. Huang, Y.-T. and Pan, W.-C. (2016). Hypothesis test of mediation effect in causal mediation model with high-dimensional continuous mediators. *Biometrics*, 72(2):402–413.
3. Zhao, Y., Lindquist, M. A., and Caffo, B. S. (2020). Sparse principal component based high-dimensional mediation analysis. *Computational Statistics & Data Analysis*, 142:106835.

Reviewer #2 (Remarks to the Author):

This well-written manuscript introduced an analytical pipeline for multivariate high-dimensional mediation analysis where both exposure and mediators are high-dimensional. This pipeline represents an ensemble of multiple state-of-the-art techniques and tools in variable selection, dimension reduction, and mediation testing. This proposed pipeline potentially has wide application scenarios to meet the new and emerging needs in the multi-omic analysis. As a showcase, the authors applied the pipeline to a subset of n=161 participants from a parent cohort (i.e., LIFECODES cohort) with p=38 environmental toxicants measured in urine as exposures, and q=63 endogenous biomarkers measured in plasma as mediators. The pipeline produced biologically meaningful results. Although the analysis may be underpowered given the limited sample size, and not strictly high-dimensional (p or $q \gg n$), these should not be considered as significant drawbacks- from bioinformatics and systematic biology perspective.

There are some comments to improve clarity:

1. The pairwise mediation step is disconnected in the pipeline, as the subsequential steps do not rely on the findings from the pairwise mediation test. The purpose of pairwise mediation analysis was mainly for demonstrating the sparsity of signals (which is almost always the case in reality) and the superiority of the actual high-dimensional mediation analysis (steps 2 and 3). It would be better to make step 1 as a screening step with a relaxed p-threshold to be inclusive (or consider a similar technique like sure independent screening). Then step 2 will be based on the top mediation combinations for the subsequential dimensional reduction technique. In addition, this will also be beneficial when the pipeline is applied to the real high or ultra-high dimensional settings.

Authors' Response: We thank the reviewer for their thoughtful insight and recommendations regarding the analytical steps of the proposed pipeline. On the first comment about the disconnect between steps of the pipeline: we have carefully reflected on all reviewers' suggestion and have revised our contribution to be reworded as an "Analytical framework" composed of "Approaches" rather than the previously used term "Analytical pipeline" composed of "Steps". The pairwise mediation approach may be implemented by users as a screening step depending on their specific research goals. However, in this particular presentation of analytical approaches for multivariate mediation analysis, our rationale for pairwise mediation as a stand-alone approach is due to the fact that several studies may seek to evaluate whole groups of exposures as mixtures and whole groups of mediators as biological pathways on the principal of biologically driven hypotheses. Nonetheless, we appreciate and recognize the significance of the reviewer's recommendation for a data-driven approach. As such, we have added additional text in the methods section to describe to readers that they can operationalize *Approach 1* as a screening step depending on the high-dimensional scale of biomarkers in their study:

Section 4.4, paragraph 6:

“This first pairwise mediation analysis approach represents the baseline status quo in environmental epidemiology research, given that a majority of existing studies deploy a one-at-a-time assessment of mediation pathways. This approach may be most useful when there are few exposure variables with a targeted mediation hypothesis. However, it is susceptible to biased results when multiple exposure variables are highly correlated but are analyzed separately. The bias and hence inflated type 1 error results from possible confounding due to co-exposures missing from the analytic model. Additionally, users can modify Approach 1a and utilize it as a screening step in multiple exposure settings. For example, mediation pairs that yield modest signals with relaxed significance thresholds estimated from Approach 1a may then be prioritized and passed onto Approach 2.”

2. For a robust estimation of natural mediation effects, the sequential ignorability assumptions are strong and can be easily violated, in particular, given a large number of exposures and mediators involved. The authors could consider providing some strategies (e.g., the sensitivity test provided in the mediation package) and more details to discuss this important aspect.

Authors' Response: We appreciate the reviewer's comment. We first note that the identifiability assumptions cannot be verified empirically from the observed data¹, and one can only justify the selection of confounders based on scientific knowledge. The influence of violating the identifiability assumptions can be assessed using sensitivity analysis or estimating E-values, which has been well-developed for mediation analysis^{2,3}. We've attached the figures for these sensitivity analyses below. Additionally, based on the reviewer's advice, we have added text to address these in our methods and results section:

Section 4.4, paragraph 22:

“To evaluate the robustness of our proposed analytical approach towards the assumptions that we defined earlier, including sequential ignorability, we recommend deploying two additional sensitivity analyses. First, we leverage the “medsen”'s function within the mediate package. This sensitivity analysis simulates the change in the natural indirect effect as a function of a sensitivity parameter, ρ , which is representative of unmeasured confounders influencing the variation in the mediator and outcome models⁵². Another sensitivity analysis that we recommend is the calculation of an E-value to evaluate the vulnerability of adjusted mediation models to unmeasured confounding. To calculate this parameter, we used the EValue package (version 3.0.9) and focused only on our final model of phthalate risk score and the MGE_i created from using all mediators simultaneously. In this setting, with a continuous outcome variable, we utilized the risk ratio approximation approach for standardized outcome variables⁵³. A third sensitivity analysis that can be leveraged is evaluation of the difference in mediation effect signal with the agnostic combination of all covariates in every unique possible combination. For six covariates in our study, that results in evaluation of 63 unique covariate combinations.”

Section 2.6, paragraph 1:

*“We assessed the sensitivity of our final mediation model for phthalate risk score and all mediators simultaneously using the medsens function in the mediation package. This sensitivity analysis revealed that in order to diminish the natural indirect effect, there would need to be unmeasured confounders in the respective mediator and outcome models such that the product of the variance explained (\tilde{R}^2) equals to 0.187 (**Supplemental Figure 4**). We further assessed the sensitivity of our final mediation model by calculating the E-value, which is an estimate of the magnitude of association required by unmeasured*

confounders to diminish the mediation signal we observed in our study. The E-value for our final model was 1.55 (lower bound 1.15) (**Supplemental Figure 5**). Therefore, an unmeasured confounder beyond the variables adjusted in our final model would need to have a risk ratio of 1.55 (lower bound 1.15) in association with gestational age at delivery and the mediator matrix in order to completely diminish the mediation signal that we observed in our study.”

Supplemental Figure 4. Fitted plot of the sensitivity parameter ρ (x-axis) and average causal mediation effect (y-axis) estimated from the *medsens* function in the *mediate* package (version 4.5.0). The dashed line represents the estimated average causal mediation effect when ρ equals to zero. Legend reports the product of \tilde{R}^2 when the mediation effect equals zero. Abbreviation: Natural indirect effect (NIE)

Supplemental Figure 5. Scatter plot of mediation effects (x-axis) and $-\text{Log}_{10}(\text{P-value})$ (y-axis) for all possible combination of six covariates ($n_{\text{combo}}=63$): specific gravity, maternal age, education, race, maternal BMI at initial study visit, and health insurance provider. Horizontal dashed red line indicates p-value <0.05 . Combinations with p-value <0.05 are highlighted in red, while combinations with p-value >0.05 are highlighted in grey. E-value estimation calculated for the final model in the main analysis, using the approximation of risk ratio transformation of standardized mediation effect estimate.

References:

1. Rubin, D. B. & Little, R. *Statistical Analysis with Missing Data, Vol. 793.* (2019).
2. Imai, K., Keele, L. & Yamamoto, T. Identification, Inference and Sensitivity Analysis for Causal Mediation Effects. *Statist. Sci.* **25**, 51–71 (2010).
3. Smith, L. H. & VanderWeele, T. J. Mediation E-values: Approximate Sensitivity Analysis for Unmeasured Mediator-Outcome Confounding. *Epidemiology* **30**, 835–837 (2019).

3. (Minor) All subtitles in step 3 indicate both “exposure and mediator” dimensions reduction- this could be misleading. It should be just exposure dimensions reduction.

Authors’ Response: We appreciate the reviewer’s suggestion on clarity regarding the subtitle in Approach 2 (formerly step 3). We have revised title which reads as “*Approach 2: Mediation analysis with multiple mediators*”

4. (Minor) The error term is missing in equation (3).

Authors’ Response: We thank the author for pointing out this missing term and equation (3) is now corrected.

Reviewer #3 (Remarks to the Author):

The paper conducted by Aung et al., introduced an analytical pipeline for high-dimensional mediation analysis to identify mediation pathways in the relationship between environmental toxicants and gestational age at delivery. Though the study is well-conducted, and the results are interesting, I have several major concerns.

This study included 161 women (52 cases and 109 controls) from 1600 pregnancy women who attended The LIFECODES pregnancy cohort. It is unclear what method was used to select these women. Are these selected 161 women representatives of the overall cohort population? Did the authors match the demographic characters between included and excluded populations? Why only 161 women were selected? Please clarify if the statistical power is sufficient to generate precise estimations.

Authors' Response: We appreciate the reviewer's clarification questions about the study sample that we applied our analytical framework towards. We created an internal table 1 for the reviewer and editor's reference below. First, the sample was selected based on prioritizing participants with availability of biological samples (plasma and urine) and those with the greatest number of exposure analytes measured from biobanked samples. Secondary to this was prioritizing the proportion of cases and controls as close as possible to the original larger case-control analysis in LIFECODES¹. The ratio of cases to controls in the larger study was aimed to be approximately 2:1. Recognizing that the proportion of cases in the subset (32.7%) was higher than the larger case-control analysis (27%), we applied inverse probability weights for all statistical analyses to penalize the influence of the larger proportion of cases. To clarify, this subset sample was not matched based on demographic characteristics. In regards to the reviewer's question about the representativeness, the internal table 1 below shows that some demographic and health variables are relatively close in proportion to the larger LIFECODES cohort (within 5% difference for any single category): BMI, tobacco use, alcohol use. Whereas maternal race, education, and health insurance provider were trending towards greater white and socioeconomic status in our subset sample. We recognize that there are potential biases attributable to the representativeness of the sample subset construction. Nonetheless, we discussed and applied multiple sensitivity analyses to ameliorate these biases, especially within the scope of unmeasured confounding. These sensitivity analyses included our evaluation of the covariate adjustment combinations and quantifying the robustness of findings in the presence of unmeasured confounding using the *medsens* function and E-value estimation. These results are shown below, and we discuss this in greater detail in the revised text:

Section 4.4, paragraph 22:

*“To evaluate the robustness of our proposed analytical approach towards the assumptions that we defined earlier, including sequential ignorability, we recommend deploying two additional sensitivity analyses. First, we leverage the *medsens* function within the *mediate* package. This sensitivity analysis simulates the change in the natural indirect effect as a function of a sensitivity parameter, ρ , which is representative of unmeasured confounders influencing the variation in the mediator and outcome models⁵². Another sensitivity analysis that we recommend is the calculation of an E-value to evaluate the vulnerability of adjusted mediation models to unmeasured confounding. To calculate this parameter, we used the *EValue* package (version 3.0.9) and focused only on our final model of phthalate risk score and the MGE_i created from using all mediators simultaneously. In this setting, with a continuous outcome variable, we utilized the risk ratio approximation approach for standardized outcome variables⁵³. A third sensitivity analysis that can be leveraged is evaluation of the difference in mediation effect signal with the agnostic combination of all covariates in every unique possible combination. For six covariates in our study, that results in evaluation of 63 unique combinations.”*

Section 2.6, paragraph 1:

“We assessed the sensitivity of our final mediation model for phthalate risk score and all mediators simultaneously using the *medsens* function in the *mediate* package. This sensitivity analysis revealed that in order to diminish the natural indirect effect, there would need to be unmeasured confounders in the respective mediator and outcome models such that the product of the variance explained (\tilde{R}^2) equals to 0.187 (**Supplemental Figure 4**). We further assessed the sensitivity of our final mediation model by calculating the *E*-value, which is an estimate of the magnitude of association required by unmeasured confounders to diminish the mediation signal we observed in our study. The *E*-value for our final model was 1.55 (lower bound 1.15) (**Supplemental Figure 5**). Therefore, an unmeasured confounder beyond the variables adjusted in our final model would need to have a risk ratio of 1.55 (lower bound 1.15) in association with gestational age at delivery and the mediator matrix in order to completely diminish the mediation signal that we observed in our study.”

In regard to the reviewer’s question about statistical power, we recognize that this sample is underpowered. However, the limited sample size is one major factor for creating this analytical framework as many new multi-omics studies leverage pilot studies and convenience samples to conduct preliminary research to support expansion of assays for larger epidemiologic cohorts. As we learned in this particular study, complete dimension reduction of the exposure and mediator matrix in this particular data example was sufficient and suggestively significant (although not meeting FDR cut-off) and warrants further application in larger replication studies.

Supplemental Figure 4. Fitted plot of the sensitivity parameter ρ (x-axis) and average causal mediation effect (y-axis) estimated from the *medsens* function in the *mediate* package (version 4.5.0). The dashed line represents the estimated average causal mediation effect when ρ equals to zero. Legend reports the product of \tilde{R}^2 when the mediation effect equals zero. Abbreviation: Natural indirect effect (NIE)

Supplemental Figure 5. Scatter plot of mediation effects (x-axis) and $-\text{Log}_{10}(\text{P-value})$ (y-axis) for all possible combination of six covariates ($n_{\text{combo}}=63$): specific gravity, maternal age, education, race, maternal BMI at initial study visit, and health insurance provider. Horizontal dashed red line indicates $p\text{-value} < 0.05$. Combinations with $p\text{-value} < 0.05$ are highlighted in red, while combinations with $p\text{-value} > 0.05$ are highlighted in grey. E-value estimation calculated for the final model in the main analysis, using the approximation of risk ratio transformation of standardized mediation effect estimate.

Internal Table 1: Comparison of subset sample to LIFECODES larger sample

Sample Characteristics		Subset (N=161)	Overall LIFECODES (N=482)
		Median (IQR)	Median IQR
		Count (percent)	Count (percent)
Age (years)		32.8 (4.8)	32.7 (6.7)
Gestational age at delivery (weeks)		38.7 (2.0)	39 (2.2)
Overall preterm birth			
	Case	52 (32.3%)	130 (27%)
	Control	109 (67.7%)	352 (73%)
Spontaneous preterm birth			
	Case	30 (21.6%)	75 (17.6%)
	Control	109 (78.4%)	352 (82.4%)
Initial visit BMI (median 10 weeks gestation)			
	<25 kg/m ²	83 (51.6%)	250 (51.9%)
	25-29.9 kg/m ²	46 (28.7%)	126 (26.1%)
	≥30 kg/m ²	32 (19.7%)	102 (21.2%)
	Missing	-	4 (0.8%)
Race			
	White	106 (65.7%)	282 (58.5%)
	Black	17 (10.9%)	77 (16.0%)
	Other	38 (23.4%)	123 (25.5%)
Education level			
	High school degree	18 (11.5%)	68 (14.1%)
	Technical school	15 (9.3%)	77 (16.0%)
	Junior college or some college	56 (34.6%)	139 (28.8%)
	College graduate	72 (44.6%)	187 (38.8%)
	Missing	-	11 (2.3%)
Insurance			
	Private/HMO/Self-pay	146 (90.7%)	385 (79.9%)
	Medicaid/SSI/MassHealth	15 (9.3%)	85 (17.6%)
	Missing	-	12 (2.5%)
Tobacco use			
	No smoking during pregnancy	152 (94.6%)	445 (92.3%)
	Smoked during pregnancy	9 (5.4%)	31 (6.4%)
	Missing	-	6 (1.2%)
Alcohol use			
	No alcohol use during pregnancy	153 (95.0%)	452 (93.8%)
	Alcohol use during pregnancy	8 (5.0%)	20 (4.1%)
	Missing	-	10 (2.1%)
Fetal sex			
	Female	75 (46.3%)	213 (44.4%)
	Male	86 (53.7%)	267 (55.6%)

References:

1. Ferguson, K. K., McElrath, T. F. & Meeker, J. D. Environmental phthalate exposure and preterm birth. *JAMA Pediatr* **168**, 61–67 (2014).

“Maternal urine samples were collected at median 26 weeks gestation”. Why urine samples during other stages of pregnancy were not collected (or used)? Plenty of previous studies have shown high variability of urinary phthalate metabolites, metals, polycyclic aromatic hydrocarbon metabolites, and phenolic analytes among pregnant women. Unless the authors have ascertained that these exposure biomarkers are reliable during pregnancy, I don’t believe a single measurement can tell the true association between exposure and adverse birth outcomes based on such a small sample size. Though the authors mentioned this point as a limitation of this study, more data is needed to support why this should or shouldn’t be a concern.

Authors’ Response: We thank the reviewer for presenting these important questions. We wholly admit that that we were limited to single time point measurements of biomarkers for this study due to funding limitations and laboratory capacity. Recognizing that we had limited ability to measure assays of the mediators in limited samples, we selected the study visit with time point median 26 weeks gestation for two chief reasons, with the goal of expanding measurement to all study visits with future funding. The first reason for selecting this time point is that towards the later pregnancy study visit there is some significant decline in participants given that several give birth preterm and therefore there is biased missingness due to delivery prior to 37 weeks gestation. The second reason is that our research group previously published studies evaluating the associations between phthalates and preterm birth using all four study visits¹, and evaluating the study visits cross-sectionally². In evaluating both of these studies, we learned that the single time point analysis of phthalates at mediation 26 weeks gestation not only contributed the greatest towards the overall repeated measures analysis, but also yielded comparable effect estimates. These data provide evidence in support that our particular study design for mediation analysis at this time point in pregnancy is reliable. However, we recognize that expansion data is important in the future. This will also require the development of novel statistical methods for longitudinal high-dimensional mediation analysis with repeated measurements – which we are very enthusiastic to pursue in the future, however, it is a statistical methods development that is beyond the scope of what this present study is attempting to provide to the scientific community. Particularly, many multi-omics studies are limited to and begin with single time point measurements of assays, therefore we genuinely believe this framework has wide applications for existing and emerging pilot studies. We added additional text in the limitations discussion:

Discussion section, paragraph 7:

“The cross-sectional nature of our study is also vulnerable to exposure measurement misclassification given that environmental toxicants such as phthalates, phenols, and polycyclic aromatic hydrocarbons have short half-lives in humans. However, when we previously investigated phthalates and preterm birth using four time points¹⁵ and evaluated single time points cross-sectionally³⁹, we observed consistent associations between the repeated measures analysis and the time point that was used for this present study (26 weeks gestation).”

References:

1. Ferguson, K. K., McElrath, T. F. & Meeker, J. D. Environmental phthalate exposure and preterm birth. *JAMA Pediatr* **168**, 61–67 (2014).

- Ferguson, K. K., McElrath, T. F., Ko, Y.-A., Mukherjee, B. & Meeker, J. D. Variability in urinary phthalate metabolite levels across pregnancy and sensitive windows of exposure for the risk of preterm birth. *Environment International* **70**, 118–124 (2014).

I guess the blood sample was also collected at a single point in time from each woman. Similar to my concern raised above, did you check the reproducibility of 63 endogenous biomarkers? Did the authors ascertain that a single measurement is reliable to reflect the levels of these biomarkers over a period of days, weeks, or months? Why blood and urine samples collected at median 26 weeks gestation? Early and late pregnancies are also critical for fetal growth.

Authors' Response: We thank the reviewer for raising these questions about the measurement of endogenous biomarkers in our study. Within the context of limitations for one-time sampling of biomarkers, we can also look to previous studies that have assessed biomarkers in repeated measurements and reported intraclass correlation coefficient (ICC). For example, one previous study of adult women (n=9) measured multiple blood lipid biomarkers (13-HODE, 9-HODE, 13-HpODE, 9-HpODE, and linoleic acid) in 8 repeated measurements throughout a menstrual cycle¹. In this study, Browne and colleagues (2007) observed high ICC values for each of these biomarkers, ranging from 0.81 – 0.97, indicating that these particular lipid biomarkers had low intra-individual variability. We also previously evaluated repeated measurements of inflammatory markers (cytokines) and found high ICC values (>0.75)². These results provide preliminary evidence that single time point measurements may have good utility in some study samples. Furthermore, this also underlines a major gap in the literature. In regard to the reviewer's comment about other periods of gestation that are critical for fetal growth, we recognize the importance for evaluating this particular research question about longitudinal associations. This approach will require a new methodological framework of newly developed statistical methods for high-dimensional multivariate mediation analysis with repeated measurements of exposures and mediators, which is beyond the scope of the current study. Our intention in this study was to provide a flexible framework that can be used by future investigators for single time point measurements. We previously investigated the predictive capacity of eicosanoids to classify preterm birth using these single time point measurements and observed moderately reliable predictive capacity, indicating that this time point of median 26 weeks gestation is an important time point³. Future studies should seek to evaluate the predictive capacity of biomarker combinations during specific windows of gestation for preterm birth and fetal growth. We have added additional text in the limitations to discuss this:

Discussion section, paragraph 7:

“Single time point measurements of endogenous biomarkers further limits inference of long-term biological effects. However, we can evaluate previous repeated measures studies of endogenous biomarkers and assess their intra-class correlation coefficients to quantify the reliability of single measurements. For example, a previous repeated measures study of select eicosanoids in women (n=9) observed low intra-individual variability across eight repeated blood measurements⁴⁰. In the LIFECODES cohort, we have also leveraged repeated measurements of the inflammatory biomarkers (CRP and cytokines), and observed moderate intra-individual variability¹⁰. These results provide some indication that although single time point measurements of eicosanoids and inflammatory markers are not ideal, they may have moderately good utility to assess biological effects. Future studies should build upon these findings to investigate environmental exposures and cytochrome p450 products as mediators in a longitudinal design in order to evaluate other vulnerable windows of development during pregnancy.”

References:

1. Browne, R. W. *et al.* Analytical and biological variation of biomarkers of oxidative stress during the menstrual cycle. *Biomarkers* **13**, 160–183 (2008).
2. Ferguson, K. K., McElrath, T. F., Chen, Y.-H., Mukherjee, B. & Meeker, J. D. Longitudinal Profiling of Inflammatory Cytokines and C-reactive Protein during Uncomplicated and Preterm Pregnancy. *Am J Reprod Immunol* **72**, 326–336 (2014).
3. Aung, M. T. *et al.* Prediction and associations of preterm birth and its subtypes with eicosanoid enzymatic pathways and inflammatory markers. *Scientific Reports* **9**, 17049–17 (2019).

Reviewers' Comments:

Reviewer #1:

Remarks to the Author:

Thank you for a thorough revision. Almost all concerns I had have been addressed. The paper is much improved and the presentation of the analytical pipeline is more clear. The following are some minor points about the presentation and illustration of the results.

1. Figure 5. The effects of Metal Risk Score → Parent Compound → Gestational Age at Delivery path are missing. May need to demonstrate the reason both in the text and in the figure caption. The results of Phenol Risk Score with all Biomarkers as the mediator seem inconsistent with the results of individual mediator group. For example, the total effect and NDE are all positive and significant in the individual mediator group analysis; while they are insignificant in the All Biomarkers analysis. Can the authors discuss the potential reasons of this inconsistency? Another one is the NIE of Phthalate Risk Score, which is negative and significant with All Biomarkers as the mediator; while only significant in the Cytochrome P450 Pathway. The NIE of Cyclooxygenase and Parent Compound Pathways are almost zero. This is a bit counterintuitive, the authors may elaborate more. A minor point about the figure, on the y-axis labeling, should it be "Parent Compound Pathway" instead of "Parent Compound" to be consistent with the rest? Another question is why this figure does not include the Inflammatory mediator group?

2. The report of mediator group results seems not very consistent across figure and tables. For example, Figure 4 reported 5 mediators groups plus All Biomarkers; while in Figure 5, 4 groups plus All Biomarkers are presented. Table 2 shows 7 groups, and Table 3, again 5 groups plus All Biomarkers.

Reviewer #2:

Remarks to the Author:

The authors have sufficiently addressed all my concerns.

Reviewer #3:

Remarks to the Author:

The authors have addressed all of my concerns.

One last point, the internal table 1 should be showed in supporting data, and described/discussed accordingly.

NCOMMS-20-23029
Response to reviewers

Title: Application of a novel analytical framework for multivariate mediation analysis of environmental data

Authors: Max T. Aung¹, Yanyi Song¹, Kelly K. Ferguson², David E. Cantonwine³, Lixia Zeng⁴, Thomas F. McElrath³, Subramaniam Pennathur^{4,5,6}, John D. Meeker⁷, Bhramar Mukherjee^{1,8}

REVIEWERS' COMMENTS

Reviewer #1 (Remarks to the Author):

Thank you for a thorough revision. Almost all concerns I had have been addressed. The paper is much improved and the presentation of the analytical pipeline is more clear. The following are some minor points about the presentation and illustration of the results.

1. Figure 5. The effects of Metal Risk Score → Parent Compound → Gestational Age at Delivery path are missing. May need to demonstrate the reason both in the text and in the figure caption. The results of Phenol Risk Score with all Biomarkers as the mediator seem inconsistent with the results of individual mediator group. For example, the total effect and NDE are all positive and significant in the individual mediator group analysis; while they are insignificant in the All Biomarkers analysis. Can the authors discuss the potential reasons of this inconsistency? Another one is the NIE of Phthalate Risk Score, which is negative and significant with All Biomarkers as the mediator; while only significant in the Cytochrome P450 Pathway. The NIE of Cyclooxygenase and Parent Compound Pathways are almost zero. This is a bit counterintuitive, the authors may elaborate more. A minor point about the figure, on the y-axis labeling, should it be “Parent Compound Pathway” instead of “Parent Compound” to be consistent with the rest? Another question is why this figure does not include the Inflammatory mediator group?

Authors' Response: We appreciate the reviewer's careful reading of our manuscript, the follow up questions and the opportunity to clarify our results and interpretations.

In regard to the first point about the metal risk score and lack of estimates when evaluating mediation for parent compound biomarkers: the population value decomposition method did not estimate a mediator group effect vector of coefficients for parent compounds in relation to metal risk score and gestational age at delivery. This suggests that the parent compound biomarkers contribute either null or sparse effects when maximizing the likelihood of structural equation modeling step in the population value decomposition method. This is also consistent with observations of low correlations ($-0.18 < \rho < 0.17$) between metals and parent compounds (exposure to mediator associations). We interpret this as an overall lack of the population value decomposition method to detect a mediation signal and construct directions of mediation (mediator group effect [MGE]), therefore we were unable to build a mediation model using the *mediate* package. This explanation is also relevant for the reviewer's last comment about why the figure does not include the inflammatory mediator group. We observed low correlation ($\rho < 0.3$) between almost all of the exposures and inflammatory markers, with the exception of one phthalate metabolite MECPP and the cytokine TNF- α ($\rho = 0.45$). Therefore, the mediator group effect for inflammatory markers was not estimated by population value decomposition, underlining null or sparse

mediation effect in the maximization of the likelihood of structural equation modeling step. We have added a new supplemental table 5 in the excel file reporting these correlation coefficients. Additionally, both of these points were added into the Figure 5 caption to explain the absence:

“In cases where population value decomposition did not produce an estimate of a mediator group effect (parent compounds and metal risk score as well as the inflammatory markers group effect for all risk scores) the results are absent from the Figure. This indicates that the mediation signal was sparse for those particular pairs of exposures and mediators.”

And further explained in the Results section, page 7, paragraph 3:

“Inflammatory markers group effect could not be estimated by population value decomposition, indicating either sparse or null mediation signals across all four environmental risk scores. The fact that mediator group effects were not estimated by the population value decomposition algorithm is partly due to the bivariate correlation structure between exposures and mediators. We observed low correlation ($\rho < 0.3$) between almost all of the exposures and inflammatory markers, with the exception of one phthalate metabolite MECPP and TNF- α ($\rho = 0.45$) (Supplemental Table 5). Similarly, the parent compound mediator group effect could not be estimated by population value decomposition for the metal risk score. This is also consistent with observations of low correlations ($-0.18 < \rho < 0.17$) between metals and parent compounds (Supplemental Table 5).”

In regard to the reviewer’s second point about explaining the phenol risk score results in Figure 5, the primary reason why the “All biomarkers” results are positive but not significant for the total, natural indirect effect (NIE), and natural direct effect (NDE) estimates is due to the fact that All biomarkers mediator group effect includes inflammatory markers, oxidative stress markers, and protein damage markers. Likely, these biomarkers are dampening the total effect signal that appears otherwise significant and positive in the cyclooxygenase, cytochrome p450, lipoxygenase, and parent compound mediator group effect results. We have added additional text in the results section to explain this:

Page 8, paragraph 2:

“The phenol risk score exhibited only significant positive total and direct effects in association with gestational age at delivery in the grouped analyses for cyclooxygenase pathway, cytochrome p450 pathway, lipoxygenase pathway, and parent compounds (Figure 5). The corresponding mediation effects were not significant. When evaluating all biomarkers simultaneously, the phenol risk score’s total, and direct effects were no longer significant, albeit these effects alongside the mediation effect were suggestive (Figure 5). A plausible explanation lies in the fact that analysis with all biomarkers simultaneously, includes inflammatory, protein damage, and oxidative stress markers, which collectively diminished the total and direct effects observed in subgroup analysis of eicosanoid enzymatic pathway groups and parent compounds.”

In regard to the reviewer’s third point about the phthalate risk score: the fact that All biomarkers mediator group effect exhibits significant total, NIE, and NDE estimates while among the individual mediator groups only cytochrome p450 pathway exhibits such effects indicates that the metabolites belonging to the cytochrome p450 contribute the greatest to the All biomarkers mediation signal. These findings are also biologically consistent with previous work where we observed that the cytochrome p450 pathway eicosanoids yielded the greatest predictive capacity for preterm birth and spontaneous preterm birth, more so than any of the other enzymatic pathways (cyclooxygenase, lipoxygenase, and parent compound) (Aung et al. 2019). We have added additional text in the discussion section to explain this:

Page 12, paragraph 2:

“Importantly, the phthalate risk score was associated with decreased gestational age at delivery and this effect was mediated by all of the mediators simultaneously. As we disaggregated the mediators into groups, we learned that this mediation effect was largely driven by cytochrome p450 derived eicosanoids. This finding is also biologically consistent with previous work where we observed that the cytochrome p450 pathway eicosanoids yielded the greatest predictive capacity for preterm birth and spontaneous preterm birth, more so than any of the other mediator groups (cyclooxygenase, lipoxigenase, and parent compound)¹³”

For the fourth minor comment, given that parent compounds are not belonging to a specific enzymatic pathway (which is the case with eicosanoid metabolites), we think that they should remain being described as parent compound.

References:

Aung, M. T. *et al.* Prediction and associations of preterm birth and its subtypes with eicosanoid enzymatic pathways and inflammatory markers. *Scientific Reports* 1–17 (2019). doi:10.1038/s41598-019-53448-z

2. The report of mediator group results seems not very consistent across figure and tables. For example, Figure 4 reported 5 mediators groups plus All Biomarkers; while in Figure 5, 4 groups plus All Biomarkers are presented. Table 2 shows 7 groups, and Table 3, again 5 groups plus All Biomarkers.

Authors' Response: We thank the reviewer for the opportunity to clarify the representation of our results across figures. The differences in mediator groups showcased across figures is due to the fact that pairwise mediation analysis of single pollutants and single mediators were performed across all mediators (Figure 4); whereas for the dimension reduction *Approach 2*, as suggested by one of the reviewers in the last revision, we opted to exclude mediator groups with less than 5 biomarkers (e.g. oxidative stress and protein damage markers). Therefore, Figure 4 does not contain those two groups and estimated a mediation signal for the inflammatory markers when evaluating individual toxicants. In Figure 5, we learned that from using the risk scores, we did not observe mediation signals – via population value decomposition – for the inflammatory markers mediator group which is why that group is not illustrated in the forest plot. Table 2 relies on pathway lasso results, which technically evaluated all biomarkers simultaneously, and the organization used in the table was to help the reader quickly and clearly see which biomarkers (according to mediator groups) were detected by the pathway lasso method. The alternative would be to show a running list of the biomarkers, which may be difficult for readers to delineate the biological significance for comparing across methods. Finally, for Table 3, using the *hima* method, there is an option to group in addition to using “All biomarkers” simultaneously, therefore we followed the same principle that we applied with population value decomposition where we only used mediator groups with greater than 5 biomarkers. Recognizing that these nuances are important, we have added additional text in the notations for each Figure and Table:

Figure 4: *“Mediator groups with less than five biomarkers (oxidative stress and protein damage) were omitted from population value decomposition.”*

Figure 5: *“In cases where population value decomposition did not estimate a mediator group effect (parent compounds and metal risk score as well as the inflammatory markers group effect for all risk scores), this indicates that the mediation signal was sparse for those particular pairs of exposures and mediators.”*

Table 2: *“Pathway lasso implemented all biomarkers simultaneously, therefore the categorization of individual mediators into mediator groups are chiefly for biological relevance, and not statistically unique models.”*

Table 3: *“Mediator groups with less than five biomarkers (oxidative stress and protein damage) were omitted from group-specific analysis using hima”*

Reviewer #2 (Remarks to the Author):

The authors have sufficiently addressed all my concerns.

Authors' Response: We thank the reviewer for reviewing our revised manuscript.

Reviewer #3 (Remarks to the Author):

The authors have addressed all of my concerns.

One last point, the internal table 1 should be showed in supporting data, and described/discussed accordingly.

Authors' Response: We thank the reviewer for their suggestion and have added internal table 1 as supplemental table 1 in the supplemental tables excel spreadsheet. We have also added additional text in the results section:

Page 4, paragraph 3:

“Sample characteristics were also compared to the larger parent LIFECODES cohort in Supplemental Table 1. Some demographic and health variables are relatively close in proportion to the larger LIFECODES cohort (within 5% difference for any single category): BMI, tobacco use, alcohol use (Supplemental Table 1). However, maternal race, education, and health insurance provider were trending towards greater white and socioeconomic status in the current subset sample compared to the larger LIFECODES cohort (Supplemental Table 1).”